



# Wind tunnel experiments on wind turbine wakes in yaw: Redefining the wake width

Jannik Schottler[1], Jan Bartl[2], Franz Mühle[3], Lars Sætran[2], Joachim Peinke[1,4], and Michael Hölling[1]

[1]ForWind, University of Oldenburg, Institute of Physics, Oldenburg, Germany
[2]Department of Energy and Process Engineering, Norwegian University of Science and Technology, Trondheim, Norway
[3]Faculty of Environmental Sciences and Natural Resource Management, Norwegian University of Life Sciences, Ås, Norway
[4]Fraunhofer IWES, Oldenburg, Germany

*Correspondence to:* Jannik Schottler (jannik.schottler@forwind.de)

**Abstract.** This paper presents an investigation of wakes behind model wind turbines, including cases of yaw misalignment. Two different turbines were used and their wakes are compared, isolating effects of boundary conditions and turbine specifications. Laser Doppler Anemometry was used to scan a full plane of a wake normal to the main flow direction, 6 rotor diameters downstream of the respective turbine. The wakes of both turbines are compared in terms of the time averaged main flow component, the turbulent kinetic energy and the distribution of velocity increments. The shape of the velocity increments' distributions is quantified by the shape parameter $\lambda^2$.

The results show that areas of strongly heavy-tailed distributed velocity increments are surrounding the velocity deficit in all cases examined. Thus, a wake is significantly wider when two-point statistics are included as opposed to a description limited to one-point quantities. As non-Gaussian distributions of velocity increments affect loads of downstream rotors, our findings impact the application of active wake steering through yaw misalignment as well as wind farm layout optimizations and should therefore be considered in future wake studies, wind farm layout and farm control approaches. Further, the velocity deficits behind both turbines are deformed to a kidney-like curled shape during yaw misalignment, for which parameterization methods are introduced. Moreover, the lateral wake deflection during yaw misalignment is investigated.

## 1 Introduction

Due to the installation of wind turbines in wind farm arrangements, the turbine wakes become inflow conditions of downstream rotors, causing *wake effects*. Those include a reduced wind velocity and increased turbulence levels. The former cause power losses of up to 20% (Barthelmie et al., 2010) in wind farms, while the latter are linked to increased loads of downstream turbines, affecting fatigue and life time (Burton et al., 2001). In order to mitigate wake effects, various concepts of active wake control strategies have been proposed and investigated. One concept is an active wake steering by an intentional yaw misalignment, where the velocity deficit behind a rotor is deflected laterally by misaligning it with the mean inflow direction. The possibility of wake re-direction by yawing was observed and investigated by means numeric simulations (e.g. Jiménez et al., 2010; Fleming et al., 2014b), in wind tunnel experiments (e.g. Medici and Alfredsson, 2006; Campagnolo et al., 2016) and in full-scale field measurements by Trujillo et al. (2016). Further, the potential of increasing the power yield in a wind farm





configuration was explored experimentally (Schottler et al., 2016), numerically (e.g. Fleming et al., 2014b; Gebraad et al., 2014) and in a field test in a full-scale wind farm (Fleming et al., 2017), showing promising results as the total power yield could be increased in the mentioned studies.

As the applicability of the concept to future wind farms require a thorough understanding of the wakes behind yawed wind turbines, this study examines the wakes behind model wind turbines during yaw misalignment. Experimental studies are necessary to validate numeric results, to tune engineering models and to gain a deeper understanding of the present effects in a controlled laboratory environment. However, when examining wake effects experimentally, varying turbine models are used. Those models strongly differ in their complexity and design, including blade design, geometry or control concepts. The simplest model is a drag disc concept, where a wind turbine is modeled by a porous disk in the flow as done by España et al. (2012) or Howland et al. (2016). Moreover, rotating turbine models have been used in numerous studies, where the design and complexity of the models vary significantly. Examples include (Medici and Alfredsson, 2006), (Bottasso et al., 2014), (Abdulrahim et al., 2015), (Rockel et al., 2016) or (Bastankhah and Porté-Agel, 2016). In contrast to numerical studies, where the vast majority of the research community uses consistent turbine models (NREL 5 MW (Jonkman et al., 2009) or DTU 10 MW (Bak et al., 2013) reference turbines for example), experiments lack certain systematics and comparability due to varying turbine models, facilities and measurement techniques. The present study aims to compare the wakes of two different model wind turbines in the same facility, using comparable boundary conditions as far as possible. Therewith, a separation between general wake effects and turbine specific observations can be achieved.

We present wake analyses ranging from mean quantities to higher order statistics. Average mean flow components are of relevance when assessing the energy yield of potential downstream turbines. An investigation of turbulence parameters such as the turbulent kinetic energy (TKE) is linked to fluctuating inflow conditions, which is important for loads of downstream turbines and therewith their lifetime (Burton et al., 2001). To gain a deeper insight, we extend our analyses to two-point statistics. More precisely, velocity increments are analyzed, allowing for a scale dependent analysis of flows. Non-Gaussianity of the distributions of velocity increments has been reported not only in small scale turbulence (Frisch, 1995), but also in the atmospheric boundary layer (e.g. Boettcher et al., 2003; Liu et al., 2010; Morales et al., 2012). To what extent statistical characteristics of velocity increments are transfered to wind turbines is of current interest throughout the research community (van Kuik et al., 2016). We believed that distributions of velocity increments in wakes are of importance for potential downstream turbines as extreme events are likely to be transfered to wind turbines in terms of fluctuating loads and power output. Studies show this for a generic turbine model (Mücke et al., 2011), in a wind tunnel experiment (Schottler et al., 2017c) and by analyzing field data of a full-scale wind farm (Milan et al., 2013). Those findings make an investigation of velocity increments in wakes extremely relevant for active wake control concepts as well as for wind farm layout approaches.

This work is organized as follows. Section 2 introduces the methods used throughout the study, including the experimental methods, a concept for quantifying a wake's deflection and a definition of the examined parameters. Section 3 shows the result of the study. First, results of the non-yawed rotors are investigated and compared in Section 3.1. Wakes during yaw misalignment are analyzed in Section 3.2, including a quantification of the wake deflection. Section 4 discusses the findings



**Table 1.** Summary of main turbine characteristics. The tip speed ratio (TSR) is based on the free stream velocity $u_{ref}$ at hub height. The Reynolds number at the blade tip, $Re_{\mathrm{tip}}$, is based on the chord length at the blade tip and the effective velocity during turbine operation. The blockage corresponds to the ratio of the rotor's swept area to the wind tunnel's cross sectional area. The direction of rotation refers to observing the rotor from upstream, with (c)cw meaning (counter)clockwise. The thrust coefficients were measured at $\gamma = 0°$.

| Turbine | Rotor diameter | Hub diameter | Blockage | TSR | $Re_{\mathrm{tip}}$ | Rotation | $c_T$ |
|---------|---------------|--------------|----------|-----|---------------------|----------|-------|
| ForWind | 0.580 m | 0.077 m | 5.4 % | 6 | $\approx 6.4 \times 10^4$ | cw | 0.97 |
| NTNU | 0.894 m | 0.090 m | 13 % | 6 | $\approx 1.1 \times 10^5$ | ccw | 0.87 |

before Section 5 summarizes this work and states the conclusions. This work is part of a joint experimental campaign by the NTNU in Trondheim and ForWind in Oldenburg. A second paper by Bartl et al. (2017) examines the influence of varying inflow conditions on the wake of one model wind turbine.

## 2 Method

### 2.1 Experimental methods

The experiments were performed in the wind tunnel of the Norwegian University of Science and Technology (NTNU) in Trondheim, Norway. The closed-loop wind tunnel has a closed test section of $2.71 \, \mathrm{m} \times 1.81 \, \mathrm{m} \times 11.15 \, \mathrm{m}$ (width $\times$ height $\times$ length). The inlet to the test section was equipped with a turbulence grid having a solidity of 35% and a mesh size of 0.24 m. Further details about the grid are described by Bartl and Sætran (2017). Two different model wind turbines were used that vary in geometry, blade design and direction of rotation. Those deliberate distinctions allow for an isolation of general effects of wake properties. The turbines will be denoted *NTNU* and *ForWind*, respectively. Table 1 summarizes the main features and differences of both turbines, further details are described by Schottler et al. (2017b). Figure 1 shows technical drawings. As can be seen, the ForWind turbine was placed on four cylindrical poles to lift the rotor above the wind tunnel boundary layer to the same hub height as the NTNU turbine, being 820 mm above the wind tunnel floor. One turbine at a time was placed on a turning table allowing for yaw misalignment, denoted by the angle $\gamma$, which is positive for a clockwise rotation of the rotor when observed from above as sketched in Figure 2. For the NTNU turbine, the reference velocity measured in the empty wind tunnel was $u_{ref,NTNU} = 10 \, ms^{-1}$ at a turbulence intensity of $TI = \sigma_u / \langle u \rangle = 0.1$. For the ForWind turbine, the inflow velocity was $u_{ref,ForWind} = 7.5 \, ms^{-1}$ and $TI = 0.05$. In both cases, the inflow was homogeneous within $\pm 6\%$ on a vertical line at the turbine's position.

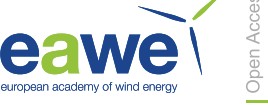
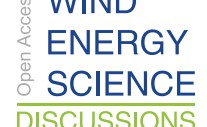


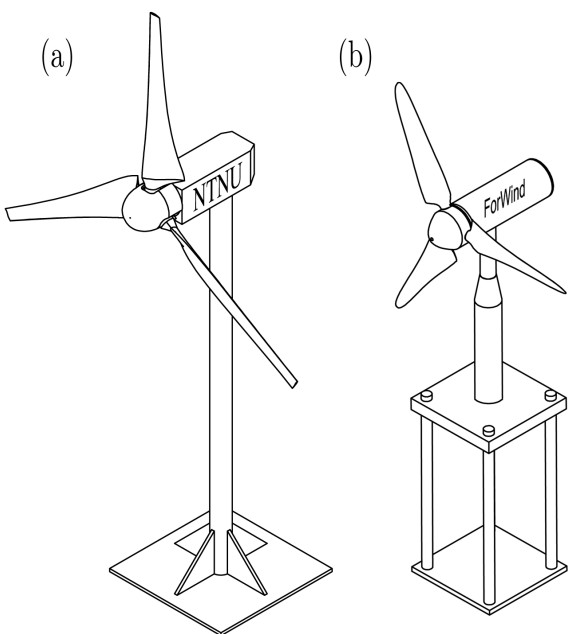

**Figure 1.** Technical drawings of the NTNU turbine (a) and the ForWind turbine (b).

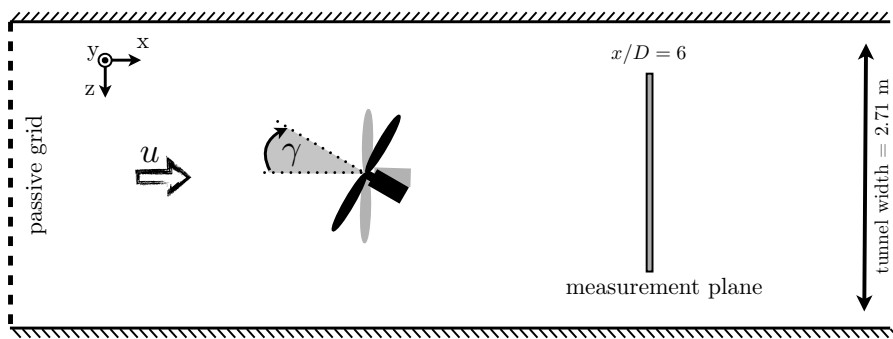

**Figure 2.** Sketch of the setup, top view. $D$ denotes the respective rotor diameter as listed in Table 1.

In this study we consider two-dimensional cuts through the wake, normal to the main flow direction at a downstream distance of $x/D = 6$ for both turbines as illustrated in Figure 2. Data were acquired using a Dantec FiberFlow two-component Laser Doppler Anemometer (LDA) system, recording the $u$- and $v$-component of the flow. The accuracy is stated to be 0.04% by the manufacturer. During turbine operation, the LDA system was traversed in the $yz$-plane, normal to the main flow direction. Each measured plane consists of 357 points, 21 in z-direction ranging from $-D$ to $+D$ and 17 points in y-direction, ranging from $-0.8D$ to $0.8D$, see Figure 3. The resulting distance separating two points of measurement is thus $0.1D$. For one location, $5 \times 10^4$ samples were recorded, resulting in time series of varying lengths of approximately 30 s. As can be seen, the NTNU



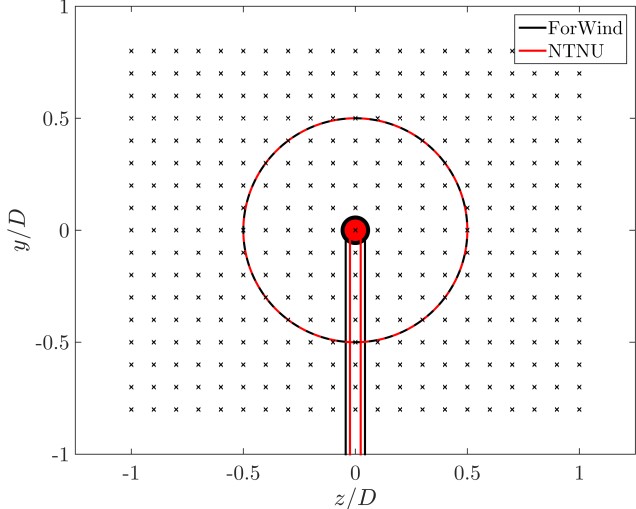

**Figure 3.** Non-dimensional measurement grid behind the rotor for $\gamma = 0°$. The respective contours of the turbines are shown in black (ForWind) and red (NTNU).

turbine has a slimmer tower and nacelle relative to its rotor diameter when compared to the ForWind turbine. The grid of physically measured values was interpolated to a grid of $401 \times 321 \approx 129000$ points for further analyses. The distance between the interpolated grid points is therewith reduced to $0.005\,D$. Natural neighbor interpolation is used, resulting in a smoother approximation of the distribution of data points (Amidror, 2002).

5  **2.2  Wake center detection**

In order to quantify the lateral wake position, we compute the power of a potential downstream turbine as described by Schottler et al. (2017b). A similar approach was shown by Vollmer et al. (2016). We define the potential power of a downstream turbine to be

$$P^* = \sum_{i=1}^{10} \rho A_i \langle u_i(t) \rangle^3_{A_i,t} \ . \tag{1}$$

10  The rotor area is divided in ten ring segments. $A_i$ is the area of the $i^{\text{th}}$ ring segment and $\langle u_i(t) \rangle_{A_i,t}$ denotes the temporally and spatially averaged velocity in mean flow direction within the area $A_i$. $P^*$ is estimated for 50 different hub locations in the range $-0.5\,D \le z \le 0.5\,D$, at hub height. We define the horizontal wake center as the $z$-position resulting in the minimum of $P^*$. The procedure is illustrated in Figure 4.

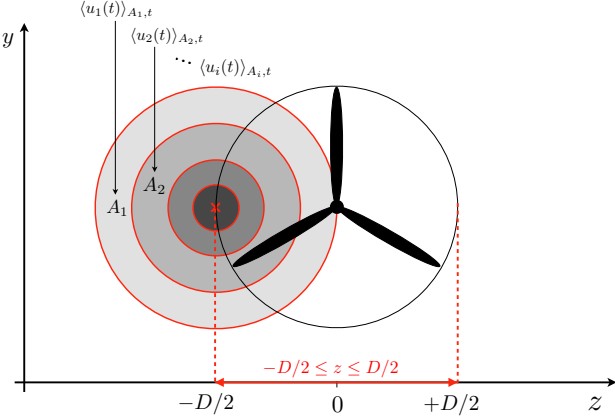

**Figure 4.** Illustration of the wake center detection method. The hub of a potential downstream turbine is located at the red $\times$. $\langle u_i(t)\rangle_{A_i,t}$ is the spatially and temporally averaged $u$-component of the velocity. The potential power $P^*$ is calculated for each ring segment and then added up. This procedure is repeated for 50 horizontal hub locations $\times$, while the position resulting in the lowest value of $P^*$ is interpreted as wake center.

## 2.3 Examined quantities

The turbulent kinetic energy (TKE) is defined by the fluctuations of the three velocity components as

$$k = 0.5 \left( \langle u'(t)^2 \rangle + \langle v'(t)^2 \rangle + \langle w'(t)^2 \rangle \right) , \tag{2}$$

where $u'(t)$ is the fluctuation around the mean of $u(t)$ so that

$$u(t) = \langle u(t) \rangle + u'(t) . \tag{3}$$

For briefness, we write $\langle u \rangle$ instead of $\langle u(t) \rangle$. As the third flow component $w$ was not recorded, we assume $w'(t) \approx v'(t)$ so that Equation (2) becomes

$$k = 0.5 \left( \langle u'(t)^2 \rangle + 2 \langle v'(t)^2 \rangle \right) , \tag{4}$$

which will be used in further analyses.

For a thorough analysis of the wake turbulence, we examine velocity changes during a time lag $\tau$ and refer to them as *velocity increments*,

$$u_\tau(t) := u(t) - u(t+\tau) . \tag{5}$$

Investigating their probability density function (PDF) allows for scale-dependent analyses of turbulent flows, including all higher order moments of $u_\tau$, hence all structure functions of order $n$, $S_\tau^n = \langle u_\tau^n \rangle$ of a velocity time series (Frisch, 1995). The
impact of certain properties of velocity increment PDFs on wind turbines is to date a widely discussed topic in wind energy




research, see (e.g. Mücke et al., 2011; Milan et al., 2013; Berg et al., 2016; Schottler et al., 2017c). For more details, we refer the reader to Morales et al. (2012) or Schottler et al. (2017c). Following Chillà et al. (1996), the shape parameter

$$\lambda^2(\tau) = \frac{ln\left(F(u_\tau)/3\right)}{4} \tag{6}$$

is used to quantify the shape of the distribution $p(u_\tau)$. $F(u_\tau)$ is the flatness of the time series of velocity increments,

$$F(u_\tau) = \frac{\langle (u_\tau - \langle u_\tau \rangle)^4 \rangle}{\langle u_\tau^2 \rangle^2} \ . \tag{7}$$

Equation (6) becomes zero for a Gaussian distribution, larger values correspond to broader, more heavy-tailed PDF. $\lambda^2$ is of practical relevance as it provides an analytical expression for the shape of $p(u_\tau)$. A discussion about the interpretation is given in Section 4. In this analysis, we compute $\lambda^2$ for time scales $\tau$ that relate to the rotor diameter $D$ of the respective turbine. Using Taylor's hypothesis of frozen turbulence (Mathieu and Scott, 2000), the length scale $r = D$ is converted to the time

scales $\tau$,

$$\tau = r/\langle u \rangle = D/\langle u \rangle \ , \tag{8}$$

whereas $\langle u \rangle$ refers to the respective time series, resulting in varying values of $\tau$ within a wake.

In order to compute $u_\tau(t)$ using Equation (5), evenly spaced data are needed. The procedure applied to uniformly re-sample the non-uniform LDA data is described in Appendix A. The approach results in a constant sampling rate for each wake.

## 3   Results

### 3.1   The non-yawed wakes

At first, we investigate wakes without yaw misalignment, $\gamma = 0°$. Figure 5 shows the contour plots of the velocity component in mean flow direction $\langle u \rangle / u_{ref}$ for both turbines, respectively, $6D$ downstream. The velocity deficits behind both turbines show a circular shape as expected, exceeding the rotor area, indicating a slight wake expansion. For both wakes, the minimum velocity

is $\langle u \rangle / u_{ref} = 0.64$. Besides those general similarities, some differences are apparent. Both graphs show the tower wake, which is pronounced stronger for the ForWind turbine. This can be explained by the larger tower diameter relative to the rotor diameter as shown in Figure 3. Similarly, the four poles the ForWind turbine is placed on (cf. Figure 1) are likely to enhance this effect. Figure 5 also reveals that the wake behind the ForWind turbine is slightly displaced vertically towards the ground. This effect can be linked to the tower wake, creating an uneven vertical transport of momentum as recently demonstrated by Pierella and

Saetran (2017). Next, the NTNU wake shows areas of velocities exceeding $\langle u \rangle / u_{ref} = 1.1$ at the edges of the velocity deficit, especially in the corners of the contour plot. Very likely, this is a blockage effect as the measurement plane is significantly larger for the NTNU turbine. This results in a higher blockage ratio (13% for the NTNU rotor, 5.4% for the ForWind rotor). As suggested by Chen and Liou (2011), blockage effects are expected for a cross-sectional blockage ratio exceeding 10% when using model wind turbines, which is confirmed here. In order to better compare both contour plots, values exceeding



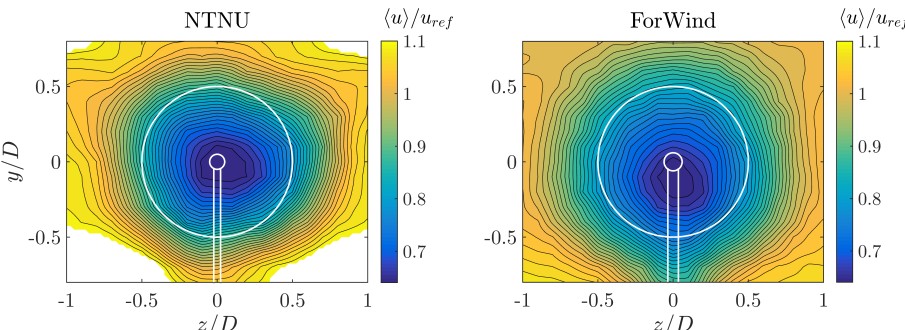

**Figure 5.** $\langle u \rangle / u_{ref}$ at $\gamma = 0°$ for the NTNU turbine (left) and ForWind turbine (right). The white lines indicate the contours of the respective turbine. Values exceeding $\langle u \rangle / u_{ref} = 1.1$ are masked.

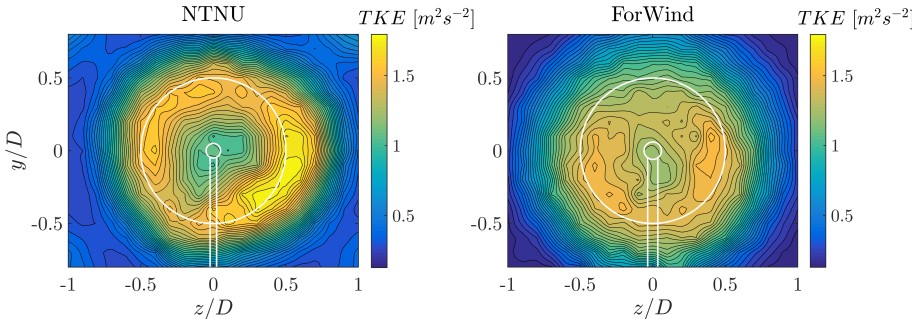

**Figure 6.** Turbulent kinetic energy (TKE) in $\mathrm{m}^2\mathrm{s}^{-2}$ according to Equation (4) for $\gamma = 0°$. Left: NTNU turbine, right: ForWind turbine.

$\langle u \rangle / u_{ref} = 1.1$ are masked.

To further analyze the wake flows, Figure 6 shows the contour plots of the turbulent kinetic energy (TKE) behind both turbines. The contours of the TKE appear as a circular shape, slightly larger than the rotor area. Behind the NTNU rotor, an outer ring of high TKE values appears more pronounced than in the center region. This observation is significantly less distinct for the

5  ForWind turbine. The differences of the pronounced ring arise most likely from the different blade geometries. The airfoil of the NTNU turbine (NREL S826) has higher lift coefficients for the relevant angles of attack and Reynolds numbers compared to the ForWind rotor (SD7003 airfoil). A comparison of both airfoils is given in Schottler et al. (2017b). As a result, larger pressure differences between suction and pressure side of the blades are expected, resulting in more pronounced tip vortices shed from the NTNU rotor. Although those are already decayed at $x/D = 6$ (Eriksen and Krogstad, 2017), the tip vortices are

10 likely to be the origin for a pronounced TKE at blade tip locations for behind the NTNU rotor.

Further increasing in complexity and completeness of the wakes' stochastic description, Figure 7 shows the contour plots of the shape parameter $\lambda^2$ behind both turbines. The length scale $\tau$ is related to the rotor diameter $D$ of the respective turbine. The scale is transfered from space to time using Taylor's Hypothesis, cf. Equation (8). In both cases, the contours of $\lambda^2$ show a circular ring, whose diameter is significantly larger than the rotor diameter. In order to quantify the qualitative shapes of the

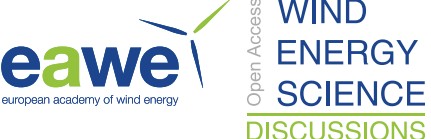

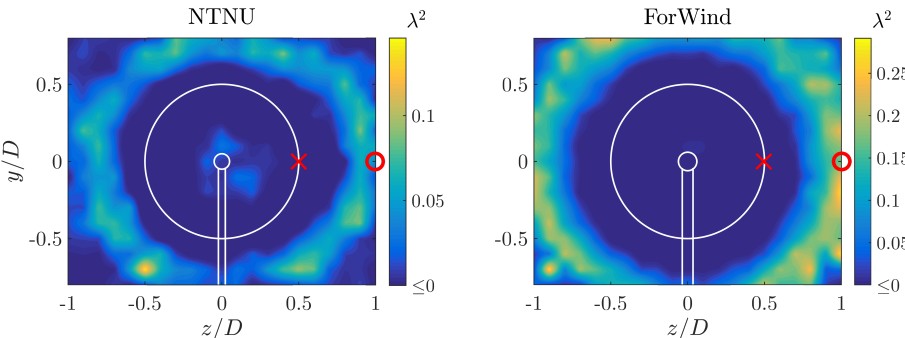

**Figure 7.** $\lambda^2$ for both turbines at $\gamma = 0°$. The time scales $\tau$ correspond to the length scale of the rotor diameter, cf. Equation (8). The red markings $\times$ and $\circ$ show measurement positions for which $p(u_\tau)$ were calculated as shown in Figure 8. Left: NTNU turbine, right: ForWind turbine. Note the different scaling.

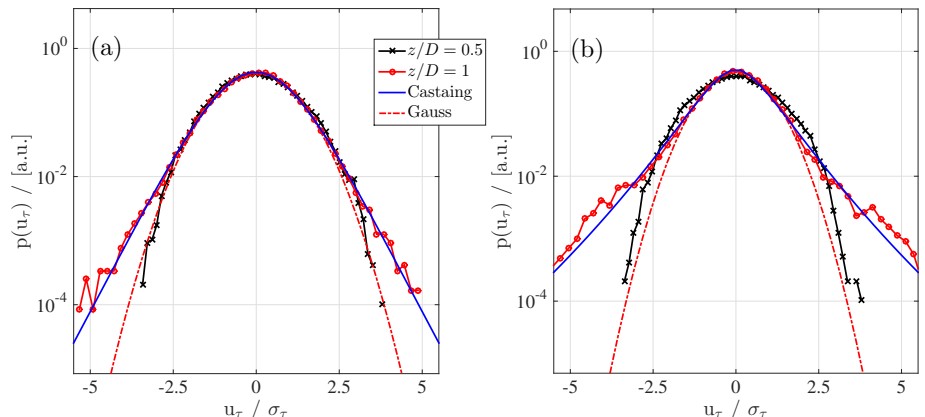

**Figure 8.** $p(u_\tau)$ of the time series at two measurement position, $(y = 0,\ z = D/2)$ and $(y = 0,\ z = D)$ corresponding to the red marks in Figure 7. (a): NTNU turbine, (b): ForWind turbine, both at $\gamma = 0°$. The time scales $\tau$ are related to the length scales of rotor diameters by Taylor's Hypothesis using Equation (8). For $z/D = 1$ (red curve) the Castaing distribution is shown with $\lambda^2_{NTNU} = 0.046$ and $\lambda^2_{ForWind} = 0.17$ (Castaing et al., 1990). A Gaussian fit is added to guide the eye.

contours shown in Figure 7, Figure 8 shows the increment PDFs of the respective time series, $p(u_\tau)$, at the positions indicated by the red marks ($\circ/\times$) in Figure 7. $u_\tau$ is normalized by the standard deviation, $\sigma_\tau$, for better visual comparison. As shown in black, the positions behind the rotor tips, where $\lambda^2 \approx 0$, reveal increment PDFs very close to a Gaussian distribution, which holds for both turbines. For $z = D$, which lies within the ring of large $\lambda^2$ values, $p(u_\tau)$ strongly deviates from a Gaussian, showing a heavy-tailed distribution, indicating more frequent occurrences of extreme events. Exemplary, in both cases an event of $5\sigma_\tau$ is underestimated by multiple orders of magnitude comparing a Gaussian distribution to the PDFs at $z = D$. Figure 8 further shows $p(u_\tau)$ based on the model proposed by Castaing et al. (1990). Those distributions were evaluated based on the $\lambda^2$ values computed by Equation (6) at $z = D$, visualizing exemplary how well the distributions' shapes are grasped by $\lambda^2$.



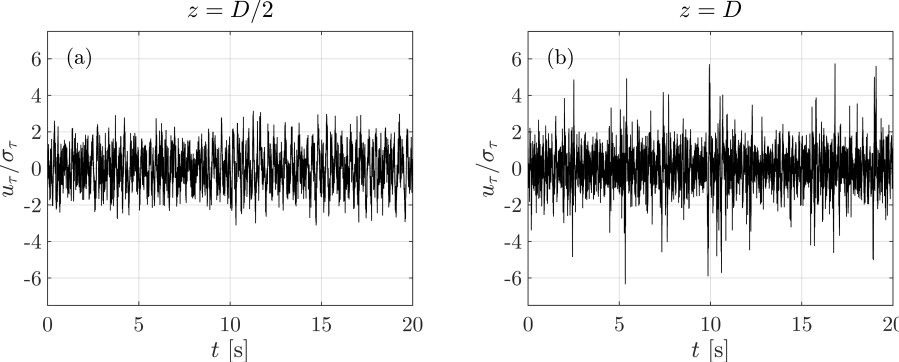

**Figure 9.** $u_\tau(t)$ at $z = D/2$ (a) and $z = D$ (b) behind the ForWind turbine, cf. Figures 7(b) and 8(b). $\sigma_\tau$ is the standard deviation of $u_\tau$.

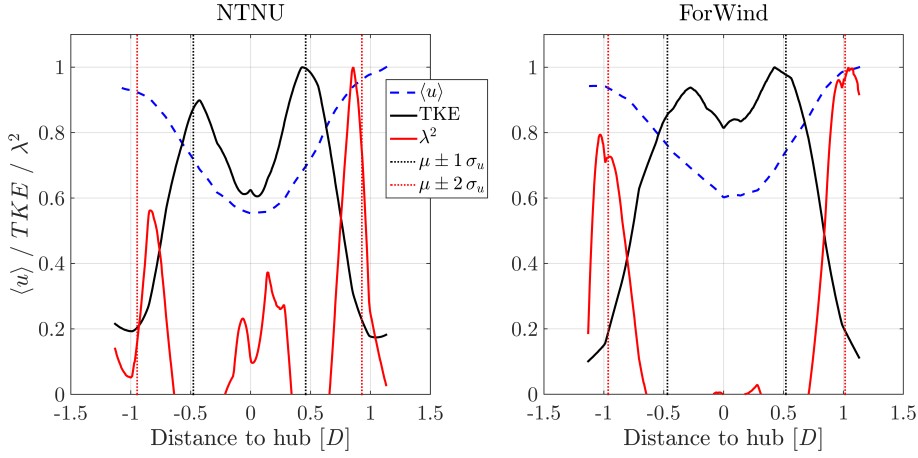

**Figure 10.** Diagonal cuts on the line $y = z$ through the contour plots for $\gamma = 0°$. Values are normalized to their respective maximum. The vertical dotted lines mark $\mu \pm 1\sigma_u$ (black) and $\mu \pm 2\sigma_u$ (red) of a Gaussian fit through the velocity deficit shown in blue.

To show the difference in $p(u_\tau)$ more intuitively, Figure 9 shows the increment time series $u_\tau(t)/\sigma_\tau$ at $z = D/2$ and $z = D$, exemplary for the ForWind turbine. It can be seen how Figure 9(a) is characterized by noisy fluctuations while Figure 9(b) shows sudden jumps e.g. extreme events.

5  Our results show that, depending on the examined quantity, different radial wake regions are of interest. To compare the varying spatial extensions of the three quantities' significant areas, Figure 10 shows diagonal cuts through the respective contour plots for the non-yawed cases along the line $y = z$. The area of pronounced TKE approximately coincides with the rotor area. The notable peaks are separated by $\approx 0.86\,D$ (NTNU) and $\approx 0.77\,D$ (ForWind), respectively, being significantly less pronounced behind the ForWind rotor as previously described. Clearly, the $\lambda^2$ peaks span a much larger distance, being approximately $1.7\,D$ (NTNU) and $2.0\,D$ (ForWind). At their location, the velocity deficit has recovered to $\geq 90\,\%$ of the free stream velocity
10  in all cases. Thus, for a thorough description of wind turbine wakes, a much larger radial area is of interest as compared





to a description restricted to mean values and the turbulent kinetic energy as often done in literature and wake models. An approximation of the lateral extension of high TKE and $\lambda^2$ values based on a Gaussian fit through the velocity deficit is given by $\mu \pm 1\sigma_u$ and $\mu \pm 2\sigma_u$, respectively, with $\mu$ being the mean value and $\sigma_u$ the standard deviation of the fit. For illustration, the dotted lines in Figure 10 mark the respective locations. It is shown that the radial areas of TKE and $\lambda^2$ can be related in this

way to the velocity deficit.

### 3.2   Wakes during yaw-misalignment

During a yaw misalignment of $\gamma = \pm 30°$, the velocity deficits behind both rotors are deflected and deformed as shown in Figure 11 by the contours of the main flow component $\langle u \rangle / u_{ref}$.

The wake is deflected sideways behind both turbines, whereas the lateral direction is dependent on the yaw angle's sign.

This is expected due to a lateral thrust component of the rotor as a result of yaw misalignment, which has been observed and described in numerous studies including (Medici and Alfredsson, 2006; Jiménez et al., 2010; Vollmer et al., 2016; Trujillo et al., 2016). The deflection of the velocity deficit is quantified using the approach described in Section 2.2, the results are listed in Table 2 including the resulting wake skew angles.

As Table 2 shows, the skew angles behind the ForWind turbine are equal apart from their sign for both directions of yaw

misalignment. The NTNU rotor however, shows slightly different deflection angles for $\gamma = 30°$ and $\gamma = -30°$, which is likely caused by blockage effects, that play a more significant role for the NTNU rotor due to the larger blockage ratio. This can also be seen in Figure 11, where speed-up effects are visible in the corners. In Schottler et al. (2017b), where the same setup was used[1], the skew angle for the NTNU rotor decreased from $x/D = 3$ to $x/D = 6$, which is a further indication for wall effects due to blockage, especially during yaw misalignment. Furthermore, both values show smaller angles as for the ForWind

turbine. In addition to the blockage effects, this is much likely caused by differences in thrust coefficient, cf. Table 1.

In Figure 11, minimum $\langle u \rangle$ values are marked, showing a vertical transport of momentum in all cases. For $\gamma = 30°$, the wake is moved upwards behind the NTNU turbines, and downwards behind the ForWind rotor. Directions are reversed for $\gamma = -30°$. Similar observations have been made by Bastankhah and Porté-Agel (2016). The vertical transport is related to an interaction of a wake's rotation with the tower shadow/ground. Our results isolate this effect, as the direction of vertical

transport is opposite comparing both turbines having an opposite direction of rotation. The fact that the vertical transport is

---

[1]In Schottler et al. (2017b), the quantification was carried out for a sheared inflow. Other aspects of the setup were equal.

**Table 2.** Wake center location as computed by the approach described in Section 2.2 with corresponding skew angles.

| Turbine | Yaw angle [°] | Wake center [$D$] | Skew angle [°] |
|---------|---------------|-------------------|----------------|
| NTNU | 30 | $-0.28$ | $\approx -2.6$ |
| NTNU | $-30$ | $0.32$ | $\approx 3.0$ |
| ForWind | 30 | $-0.38$ | $\approx -3.6$ |
| ForWind | $-30$ | $0.38$ | $\approx 3.6$ |





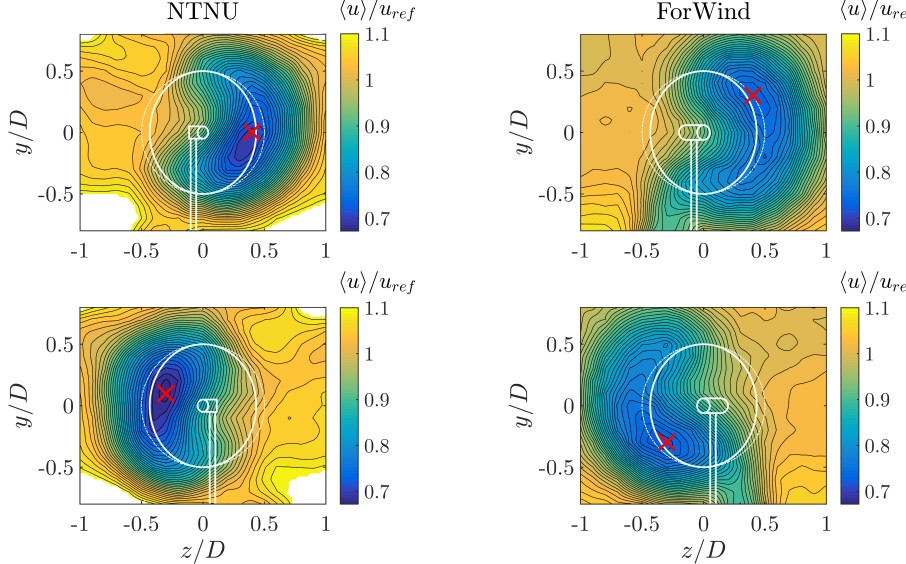

**Figure 11.** $\langle u \rangle / u_{ref}$ during yaw misalignment. Top row: $\gamma = -30°$, button row: $\gamma = 30°$. Left column: NTNU turbine, right column: ForWind turbine. The solid white lines indicate the contours of the respective turbine, while the dashed lines denote the rotor area without yaw misalignment. The red × marks the position of minimum measured velocity $\langle u \rangle$. Values exceeding 1.1 are masked for better comparison.

stronger behind the ForWind rotor further supports this explanation as the tower wake is more pronounced due to the larger tower diameter and the structure the turbines is placed on.

A deformation of the velocity deficit to a curled "kidney" shape is observed for both turbines during yaw misalignment, whereas it is slightly more pronounced behind the ForWind turbine. The curled shape behind a wind turbine model in yaw has previously 5 been observed by Howland et al. (2016) using a drag disc of 30 mm diameter and by Bastankhah and Porté-Agel (2016) using a rotating turbine model of 150 mm diameter. Figure 11 confirms these findings on two further scales. For a better comparison of the curled shape of the velocity deficit during yaw misalignment, we apply the following parametrization, exemplary shown in Figure 12(a) for the ForWind turbine at $\gamma = 30°$: data points of horizontal cuts through the wake, $\langle u \rangle_{y=\text{const.}}$, are fitted by a polynomial. The procedure is repeated for values of $y$ ranging from $-0.4\,D$ to $0.4\,D$. The positions of the polynomials' minima 10 (green marks), are fitted by a quadratic function (red line). Figure 12(b) shows the comparison of both turbines for $\gamma = \pm 30°$. As already seen in Figure 11, the wakes behind the ForWind turbine are deflected further and the curled shape is pronounced stronger, which can be attributed to the larger thrust coefficient and blockage effects. Figure 12(b) also shows that the wakes behind both turbines are slightly tilted. Looking at the black curves (ForWind turbine), an asymmetry can be noticed as the curves are tilted towards the left, while the red curves are tilted towards the right. This is illustrated by the gray, dashed lines in 15 Figure 12(b) which connect the points of intersection for $\gamma = \pm 30°$. Similar asymmetries have been observes by Bastankhah and Porté-Agel (2016) for positive and negative yaw angles, which is explained by an interaction of a wake's rotation with the tower wake and the ground. By using turbines of opposite rotation direction, we can attribute the asymmetries in vertical



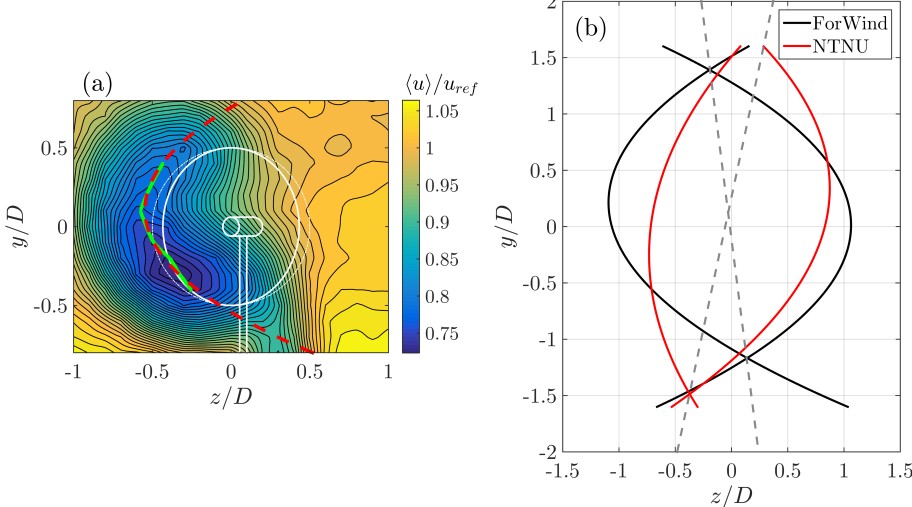

**Figure 12.** (a): Example of parameterizing the curled shape of the velocity deficit. The green markings show minimal velocities of a polynomial function used to fit the interpolated data points in a horizontal line, $y = \mathrm{const}$. The red, dashed line shows a quadratic fit based on the green markings. (b): Visualization of the curled shapes of the velocity deficits. For both turbines, the cases $\gamma = \pm 30°$ are shown. Dashed lines show a visualization of the wakes tilt, connecting the respective intersections of the curves.

transport and the tilt in opposite direction for $\gamma = \pm 30°$ to the rotation of rotor and wake. Not shown in detail here, the same effect was observed for different inflow conditions and other downstream distances, using the same setup and methods as in this study.

Adding TKE and $\lambda^2$ contours during yaw misalignment, Figure 13 shows all three examined quantities, exemplary at a yaw
5   misalignment of $\gamma = -30°$, for both turbines. The shapes of the TKE contours are deformed similarly as for $\langle u \rangle$. A curled shape evolves and the differences between both turbines as described for $\gamma = 0°$ are still notable during yaw misalignment. Similarly, the circular rings of high $\lambda^2$ values are deformed to a curled shape at $\gamma = \pm 30°$. Thus, the general effect of heavy-tailed increment PDFs surrounding the velocity deficits in a wake is stable against yaw misalignment and the resulting inflow variations at the rotor blades. Further, this finding is confirmed in Large Eddy Simulations (LES) performed at the Universidad
10   de la República, Uruguay, shown in Appendix B. Therewith, it is found to be a general effect as it is observed for all wakes considered, independent of yaw misalignment or turbine design. The red markings in Figure 13 show the approximation of the radial extension of the TKE and $\lambda^2$ based on $\mu \pm 1\sigma_u$ and $\mu \pm 2\sigma_u$. $\mu$ and $\sigma_u$ correspond to Gaussian fits of the velocity deficits at various horizontal cuts ($y$=const.) from $y/D = -0.5$ to $y/D = 0.5$. It is shown that the methods results in quite good first order approximations, also during yaw misalignment.





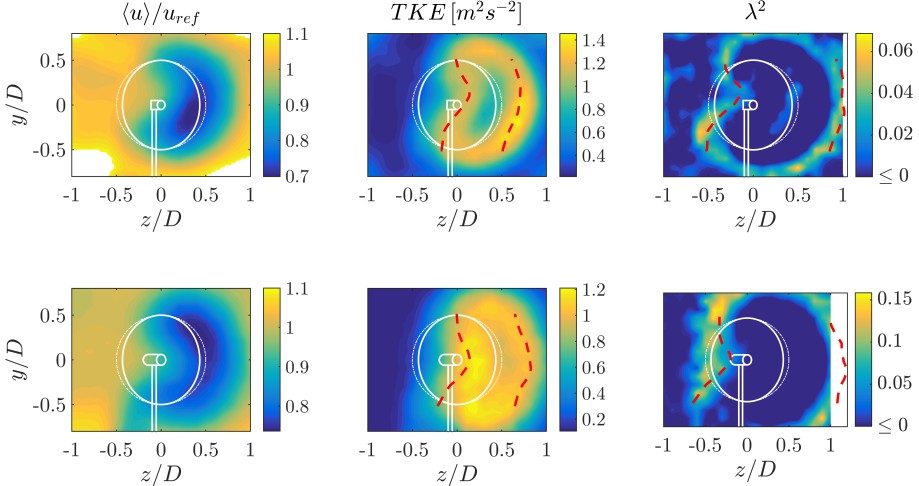

**Figure 13.** $\langle u \rangle / u_{ref}$ (left column), TKE (center column) and $\lambda^2$ (right column) for $\gamma = -30°$ behind the NTNU turbine (top row) and the ForWind turbine (bottom row). The time scale for $\lambda^2$ corresponds to the length scale of the rotor diameter. The red marks show the approximation of the respective parameter's radial extension based on $\mu \pm 1\sigma_u$ and $\mu \pm 2\sigma_u$ as described in Section 3.1.

## 4   Discussion

In this study the characterization of yawed and non-yawed wind turbine wakes is investigated and extended by taking into account a further turbulence measure, namely the intermittency parameter $\lambda^2$. We find heavy-tailed distributions of velocity increments in a ring area *surrounding* the velocity deficit and areas of high TKE in a wind turbine wake. Thus, the definition of

a wake width strongly depends on the quantities taken into account. The heavy-tailed distributions are the statistical description of large velocity changes over given time scales and are transfered to turbines in terms of loads and power output. This has been shown experimentally (Schottler et al., 2017c), numerically (Mücke et al., 2011) and in a field study by Milan et al. (2013). Consequently, our findings should be considered in wind farm layout optimization approaches, where a wake's width affects radial turbine spacing. Possibly, the ring of non-Gaussian velocity increments is a result of instable flow states, where

the flow switches between a wake and free stream state. Behind a rotor, the wake characteristics dominate the flow. Outside the wake, free stream properties are dominant. In the transition zone, a switching between both flow states is believed to result in heavy-tailed velocity increments and therewith high $\lambda^2$ values. Generally, $\lambda^2$ will be larger for smaller scales $\tau$, which is a known feature of turbulence (Frisch, 1995).

Care should be taken when interpreting $\lambda^2$ as an indicator for an increment PDF's shape. Here, we use the shape parameter as

qualitative indicator. For a more quantitative analysis, one has to consider the increment PDF of a time series directly. This is done in Figure 8 exemplary for chosen points, however, in order include all time series of a wake, using $\lambda^2$ allows for a much better visualization and comparison.





Figure 13 shows that the velocity deficit is deflected laterally during yaw misalignment, so that a potential in-line downstream turbine would exhibit a power increase as more undisturbed flow hits the rotor area at $z/D \approx -0.5$. Looking at the $\lambda^2$ contours however shows, that areas of non-Gaussian velocity increments are now deflected onto the rotor area. This becomes important when assessing the applicability of active wake steering approaches, as a gain in power has to be balanced with a potential load

increase, affecting maintenance costs and the lifetime of turbines overall.

The velocity deficit in mean flow direction $\langle u \rangle$ deforms to a curled "kidney" shape during yaw misalignment. Consequently, horizontal cuts through the wake are insufficient when characterizing wakes behind yawed rotors, resulting in misleading and incomplete conclusions when quantifying wake deflections by yaw misalignment. The parametrization of the wake's curl shown in Figure 12 should not be interpreted as quantification. Instead, we use the described approach to better compare

multiple curled wakes as done in Figure 12(b). Our analyses include the velocity deficit in mean flow direction, the turbulent kinetic energy and the shape parameter $\lambda^2$. The turbulence intensity in the wake revealed very comparable results as the TKE, which is why we restrict our analyses to the TKE.

Besides the lateral deflection, a vertical transport of the velocity deficit is observed for both turbines during yaw misalignment. Using counter-rotating turbines, this effect could be attributed to the wake's rotation and its interaction with the tower wake. In

full scale scenarios, the ground, wind shear and rotor tilt would further contribute to the effect. For potential floating turbines, a pitch motion will deflect the wake upwards, see Rockel et al. (2014). This vertical deflection will interact with the vertical transport shown in Figure 11. Consequently, the direction of yaw misalignment is believed to be of importance when applying the concept of wake steering to wind farm controls. This confirms findings by Fleming et al. (2014a) and Schottler et al. (2017a), reporting an asymmetric power output of a two-turbine case with respect to the upstream turbine's angle of yaw

misalignment.

## 5 Conclusions

This work shows an experimental investigation of wind turbine wakes, using two different model wind turbines. The analyses include the main flow component, the turbulent kinetic energy and two-point statistics of velocity increments, quantified by the shape parameter $\lambda^2$. Yaw angles of $\gamma = \{0°, \pm30°\}$ are considered at a downstream distance of $x/D = 6$.

Generally, the results of $\langle u \rangle$, the TKE and $\lambda^2$ compare well for both model turbines. Minor differences could be ascribed to the more prominent blockage (12.8% vs 5.4%) in the NTNU setup, confirming findings by Chen and Liou (2011), who state blockage effects can be neglected for a blockage ratio $\leq 10\%$.

An outer ring of heavy-tailed velocity increments surrounds the velocity deficit and areas of high TKE in a wind turbine wake. The wake features significantly non-Gaussian velocity increment distributions in those areas, where the velocity deficit

recovered nearly completely. For $\gamma = 0°$, the ring has a diameter of approximately $1.7D$ - $2D$, depending on the turbine. Based on a Gaussian fit through the velocity deficit, the radial location of intermittent increments can be approximated by $\mu \pm 2\sigma_u$, making a wake considerably wider when taking two-point statistics into account. This observation becomes important in wind farm layout optimization and active wake steering approaches through yaw misalignment.





During yaw misalignment, the circular shape of a wake is deformed to a curled kidney-shape. A method for parameterizing the curl-shape was introduced. Further, the lateral wake deflection was quantified, resulting in skew angles of $\pm 3.6°$ at $\pm 30°$ for the smaller rotor and $3.0°$ and $-2.6°$ for the larger rotor. Furthermore, vertical momentum transport in the wake during yaw misalignment was observed. The direction of vertical transport is dependent on the direction of yaw misalignment. Using

counter-rotating turbines, the effect could be attributed to an interaction of a wake's rotation with the tower wake in this study.

*Data availability.* The experimental data set is available upon request.

**Appendix A: Data preprocessing**

In order to study intermittency using the shape parameter $\lambda^2$, uniformly sampled data are needed when applying Equation (5). As the LDA measurement result in non-uniformly sampled data points, appropriate preprocessing is necessary. In the

10 following, the procedure is described that results in uniformly sampled data points. It is exemplary applied to the data of an arbitrarily chosen wake.

The time separating two samples of a time series is $\Delta t$. For one time series, $(\Delta t)^{-1}$ is plotted for all samples in Figure A1 (a). The corresponding histogram is shown in Figure A1 (b). The point corresponding to $40\,\%$ of all events is marked by the red dashed line and is referred to as $F_S$. In this example, $F_S \approx 1.17\,\mathrm{kHz}$.

This procedure is repeated for all 357 time series contained in one plane of measurement. Figure A2 shows $F_S$ for all time series, with the mean value indicated. The mean value of all $F_S$ values in one plane will be used as sampling frequency to re-sample the time series in one plane uniformly, an exemplary result is shown in Figure A3. Data points are interpolated linearly onto a vector of uniformly spaced instants defines by the new sampling rate $\langle F_S \rangle$. It should be noted that the analyses of velocity increments were performed for different constant sampling rates without showing any significant effect on the results.





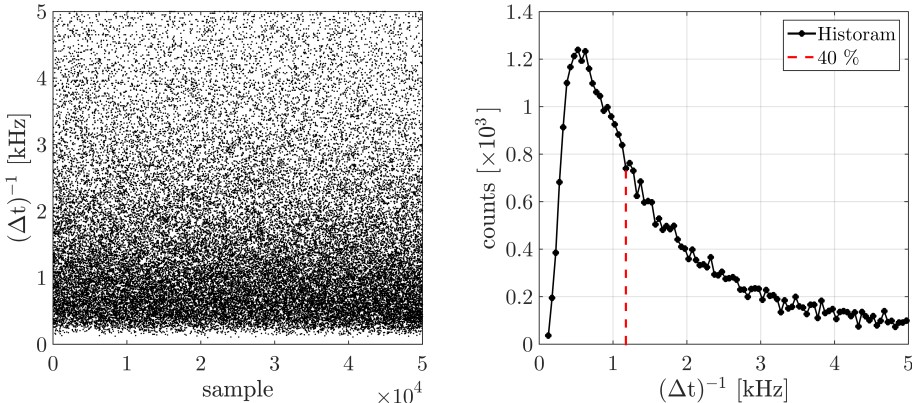

**Figure A1.** $(\Delta t)^{-1}$ for all samples (a) with the respective histogram (b), where the maximum value is marked by the red, dashed line.

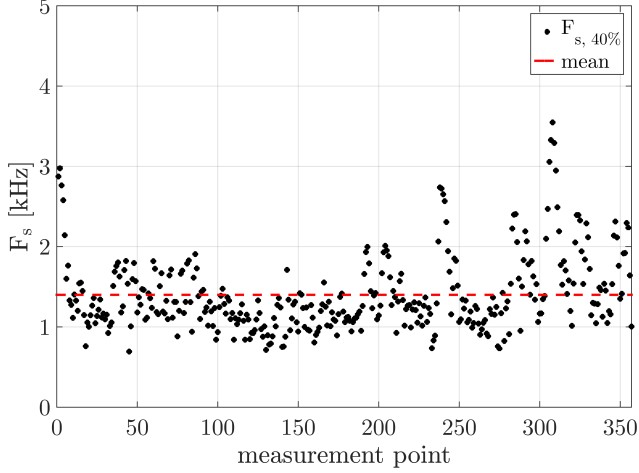

**Figure A2.** $F_S$ for all 357 time series of one wake, the mean value is indicated in red, being $\langle F_S \rangle = 1.4\,\text{kHz}$.

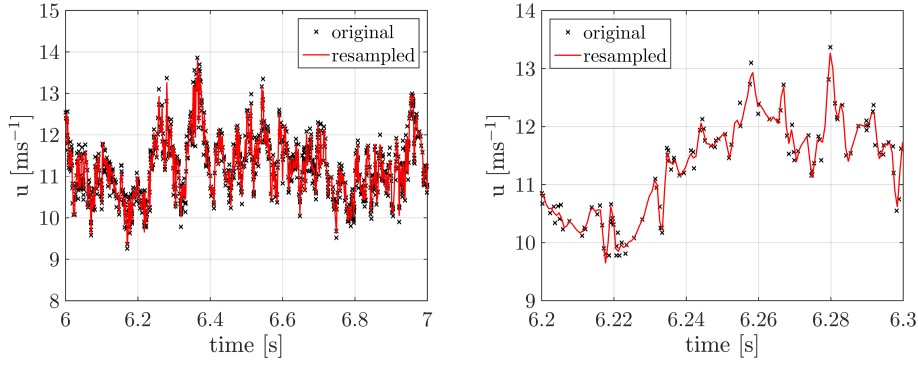

**Figure A3.** Examples of resampling the raw data $u(t)$ uniformly with $\langle F_S \rangle = 1.4\,\text{kHz}$.





## Appendix B:  LES simulations

Within the scope of the *blind test 5* project, LES simulations of the ForWind turbine in a very comparable setup were performed, where the inflow features a vertical shear as opposed to the experiments shown in this paper. The incompressible flow solver caffa3d.MBRi as described by Mendina et al. (2014) and Draper et al. (2016) was used to obtain the results shown in Figure

5   B1. The turbine was modeled by actuator lines. The top row shows $x/D = 3$, $x/D = 6$ is shown beneath. The contours of $\langle u \rangle / u_{ref}$ and $\lambda^2$ reveal very similar results compared to the experimental data. Qualitatively, it can be concluded that the outer ring of high $\lambda^2$ values and thus heavy-tailed distributions of velocity increments, that surrounds the velocity deficit of a wake, can be correctly predicted in LES simulations.



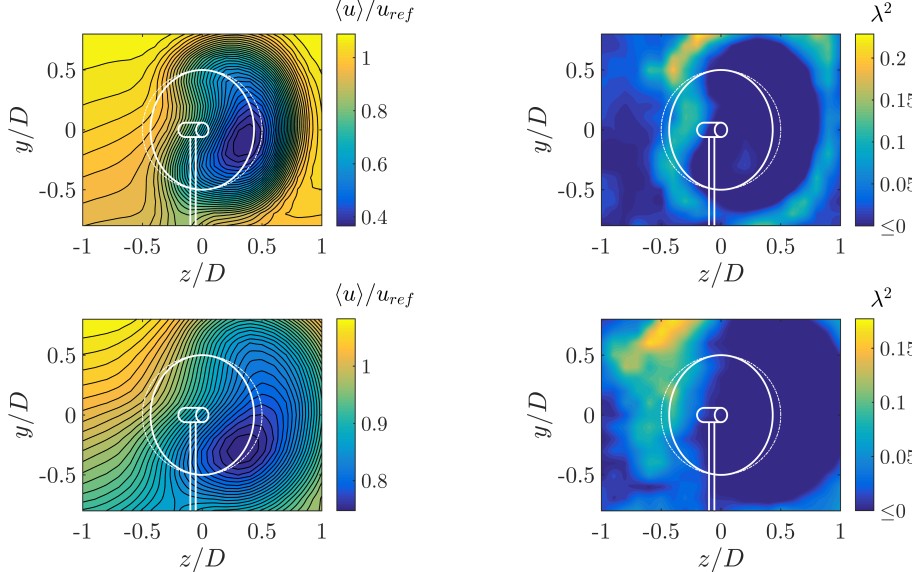

**Figure B1.** LES data of the wakes $3D$ (top row) and $6D$ (button row) behind the ForWind turbine at $\gamma = 30°$. In contrast to the experiments presented in this paper, the inflow in the LES domain features a vertical shear with comparable turbulence intensity. The time scales of $\tau$ for the $\lambda^2$ calculations correspond to the length scale of the rotor diameter.

*Competing interests.* The authors declare no competing interests.

*Acknowledgements.* The authors thank Marín Draper and Andrés Guggeri, Universidad de la República, Uruguay for performing the LES simulation and providing the data.



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
