# Peer review of "Wind tunnel experiments on wind turbine wakes in yaw: Redefining the wake width"

_Wind Energy Science, 2017_

## Referee Comment (RC1) · Anonymous Referee #1 · 30 Jan 2018

The results in this paper, together with the companion paper, provide the results of a careful wind-tunnel experiment into wind turbine wakes in yaw. Additionally, this paper provides a novel analysis of velocity increments in the flow behind turbines, and how these are effected by yawing. The results and analysis are novel and provide a useful contribution to the literature. Overall the paper is well written with good supporting figures and analysis.

Comments:

Main comment is on the impact to loads. In the introduction, and later in the paper, references to past literature documenting that their is a connection between velocity increments and loads, but the nature of the connection is not elaborated on. Could some of the findings of those papers be summarized for context? For example, are the

impacts more important for fatigue loads or extreme loads? In the companion paper, figure 11 shows a reduction in TKE during wake steering. If one is considering wake steering, to what extent would a reduction in TKE counter-balance a change in increment velocity? Is there a method to weigh these two changes? Is there a connection to loads on specific components (blades, drivetrain) or failure modes? Details in this regard would help to contextualize the findings.

Could the authors elaborate further on the connections to the companion paper. Would it make sense to bring the TKE analysis of the companion paper into this paper, and move the analysis of wake position to the companion paper? Feel free to reject this suggestion if I misunderstand the distinctions between the papers, My thinking is just that, for example, if only one of the papers dealt with estimating wake position, then this could make each of the papers more focused on specific effects. But, it would also be acceptable to further elaborate on the focus of the two papers, where they overlap and where they diverge.

Finally, the difference in rotation direction between the turbine models is very interesting. The authors use this difference to explain the asymmetries in vertical transport and tilt, could it also explain differences in displacement for positive vs negative yaw observed in the companion paper? Does the size of observed vortices vary with whether the vortex shed by misalignment is rotating in the same direction as the wake?

---

## Referee Comment (RC2) · Anonymous Referee #2 · 11 Feb 2018

The paper features a very interesting investigation, useful to the community. Well written and therefore nice to read.

Although the title clearly mentions the paper deals with a wind tunnel test, it would be good to exercise some caution in the text on the application of the results to the 'real world'.

-The reported high thrust coefficients corresponds to rather high axial induction factors towards the turbulent wake state, in how far is this representative for real life turbines nowadays and how would this affect the observed wake shapes? Has there been any attempt to clarify the effect of operational conditions on observations (partial load. full load)

[Figure]

-Blockage. Referred paper on tunnel effects refers to blockage correction (to correct freestream velocity and modify power and loads). Does the same conclusion hold for measured wake velocities or are they more sensitive to tunnel effects? Is there an influence of the asymmetry of the test section on the measured wake shape at 6D in yaw?

-2.1 pp3 Please state the cause/reason for the different TI. How was the homogeniety verified, do I understand correctly that standard deviation of flow velocity was the same in all three directions??

-2.2 pp4 motivate choice for x/D=6

-5 pp15, does blockage also depend on Ct?

-5 pp15 It is stated that another paper "Bartl, J., Mühle, F., Schottler, J., Sætran, L., Peinke, J., Adaramola, M., and Hölling, M.: Experiments on wind turbine wakes in yaw: Effects of inflow turbulence and shear, Wind Energy Science, submitted, 2017." discusses the effect of inflow TI. " Since the differences between the measurements on the 2 turbines are discussed in the conclusions, what would be the effect of the different inflow TI for the 2 campaigns on the measured differences?

---

## Referee Comment (RC3) · Anonymous Referee #3 · 14 Feb 2018

General comments

The article describes wind turbine experiments on two different model wind turbines. The results include velocity measurements on a plane, 6 rotor diameters (6D) behind the rotor, with transversal and vertical extent 2D. For each model turbine, the load cases include yaw angles of 0 and +-30deg, at a tip speed ratio of 6. The different rotor diameters and chosen test conditions lead to a ratio of about two for wind tunnel blockage and tip Reynolds number, and thrust coefficients of 0.97 and 0.87 respectively. The results are presented on the measurement plane in terms of mean velocity deficit, Turbulent Kinetic Energy (TKE) and a shape parameter describing the shape of the probability density function for velocity increments. Methods for identifying the wake center and thereby the skew angles, and wake shapes are presented. The result

include the shape parameter showing the maximum non-Gaussians behavior well outside of what we normally regard as the wake in terms of velocity deficit and TKE, and non-circular wake shapes resulting from the yaw errors.

The paper is interesting, clear, well written and easy to read. The quality of the measurements, data analysis and presentation appears to be high. The analysis provides a novel view on the wake shape and extent. The results are relevant to wind farm design, including current topics such as wake steering through manipulation of wind turbine yaw. The impact may be a broader view on what is the wake width, and strategies for wake steering, although further quantification on the impact on wind turbine loads should be carried out.

Specific comments

The anonymous referee #2 has commented on the high induction factors and the choice of position of measurement plane, the different TI, Ct and blockage ratios for the two turbines tested, and the possible impact on the measured wake velocities.

I understand that the wake effects are more easily studied at high induction factors, relatively close to the rotor, but I also share ref #2's curiosity about how this relates to real wind farms. I suggest a section showing the Ct vs. wind speed curve for a large modern wind turbine, and a few sentences about typical wind turbine spacings in recently built wind farms (along and across the main wind direction).

The anonymous referee #1 main comment is on the impact of inflow velocity increments on the loads for a wind turbine. I would like to add a few comments on this topic.

Figure 5 shows the mean velocity deficit at 6D behind the rotor. As expected, the wake (in terms of velocity deficit) has expanded somewhat, but at y/D and z/D of 1, we have more or less free stream conditions. Figure 6 shows the influence of the rotor in terms of TKE. Again we see that the wake has expanded, but at y/D and z/D of 1, we are almost at free-stream. Figure 7 is intriguing. Although the wake in terms of mean

velocity deficit and TKE is hardly present at y/D and z/D of 1, the shape parameter here shows a strong signal, close to the maximum value across the measurement plane.

My main comment is that the shape parameter can be high, but the velocity fluctuations may be too small to affect the loads. I therefore appreciate that the authors in the following figures try to present the results in different ways, but in my opinion, some more figures should be added here.

In figure 8, the probability density functions at the two points are normalized in different ways to be compared with the same Gauss distribution. What is the ratio of velocity increment standard deviations at the two positions? How would a plot look if the results were normalized in the same manner?

In figure 9, the velocity increments at the two positions are again normalized with different standard deviations. I would like to see the corresponding plots also normalized with the standard deviation at D/2.

I suggest also time series and PSD plots of longitudinal velocity fluctuations at D/2 and D, and for the free stream, all normalized in the same manner. Maybe it is better to show a comparison at D and free stream separately.

Technical corrections

Caption of Table 1, pg. 3: Is the effective velocity during turbine operation the relative wind speed with respect to the rotor tip? The blade tip of the ForWind turbine looks like it has a rounded shape. Where is the tip chord defined?

2.3, page 6: Please mention if measurements support the assumption about vertical vs transversal fluctuations. I assume you mean $<w^2>$ vs. $<v^2>$

Caption, Figure 3: Consider adding something like: For the NTNU turbine, the wind tunnel walls are located at z/D = +-3.03 and y/D = +-2.02. For the ForWind turbine, the wind tunnel walls are located at z/D = +- 4.67 and y/D = +- 3.12

Caption, Figure 11, pg. 12: Bottom row.

Caption, Figure 13: The red marks show the approximation of the respective parameter's radial extension based on $\mu \pm 1$ u and $\mu \pm 2$ u as described in Section 3.1. But I see only two red lines, is it at one or two sigma?

---

## Author Comment (AC1) · 2 Mar 2018

Authors' response to Anonymous Referee #1:

We, the authors, are very thankful for the detailed and constructive comments and greatly appreciate the willingness to review our manuscript. Please find our responses below. The original comments are shown in **bold** with the respective answers below. Excerpts of the manuscript are shown in *italic writing*, whereas additions are written in blue and deleted parts in .
Please note that the format of citations in manuscript excerpts might be changed.
Thank you very much for your efforts,

Jannik Schottler on behalf of all authors
* * *
**1)**
**Main comment is on the impact to loads. In the introduction, and later in the paper, references to past literature documenting that their is a connection between velocity increments and loads, but the nature of the connection is not elaborated on. Could some of the findings of those papers be summarized for context? For example, are the impacts more important for fatigue loads or extreme loads? In the companion paper, figure 11 shows a reduction in TKE during wake steering. If one is considering wake steering, to what extent would a reduction in TKE counter-balance a change in increment velocity? Is there a method to weigh these two changes? Is there a connection to loads on specific components (blades, drivetrain) or failure modes? Details in this regard would help to contextualize the findings.**

Thank you very much for this constructive comment. We want to answer the different aspects separately, for better clarity. Afterwards, we give some more details for completeness of the discussion.

**In the introduction, and later in the paper, references to past literature documenting that their is a connection between velocity increments and loads, but the nature of the connection is not elaborated on. Could some of the findings of those papers be summarized for context?**

To what extent intermittent characteristics of atmospheric turbulence transfer to turbine data such as torque, moments, power, etc has been investigated experimentally and numerically. Details are subject of discussion within the research community, however, relevant studies are summarized here: Milan et al. [9] analyzed power data of full scale wind turbines and of a whole wind farm, finding heavy-tailed power increments on time scales of the order seconds, suggesting intermittency is transfered from wind to power. In a wind tunnel experiment using an active grid and a model wind turbine [1], we showed that non-Gaussianity of velocity increments was transfered to power, torque and thrust data of the model turbine on the lab scale (that is the same model wind turbine as denoted *ForWind* turbine in the manuscript).
In a numeric study, Mücke et al. [10] found that intermittent flow conditions result in similarly intermittent torque increments using FAST [11] in combination with Aero-

Dyn [12]. In the manuscript, we suggest to summarize this in the introduction:

p.2, ll.25 ff:

*[...]. To what extent statistical characteristics of velocity increments are transfered to wind turbines is of current interest throughout the research community [14].*  *Schottler et al. [1] found a transfer of intermittency from wind to torque, thrust and power data in a wind tunnel experiment using a model wind turbine. Similarly, Mücke et al. [10] found a transfer of intermittency to torque data using a generic turbine model. Milan et al. [9] reported intermittent power data in a full-scale wind farm. We thus believed that distributions of velocity increments in wakes are of importance for potential downstream turbines as*  *non-Gaussian characteristics are likely to be transfered to wind turbines in terms of fluctuating loads and power output.*  *Consequently, investigations of velocity increments in wakes become extremely relevant for active wake control concepts as well as for wind farm layout approaches. A further elaboration on the connection of non-Gaussian velocity increments and loads as well of power fluctuations is given in Section 4. This work is organized as follows. [...]*

**For example, are the impacts more important for fatigue loads or extreme loads?**
Despite the above findings (intermittency is transfered to turbine data), the question remains to what extent intermittent, non-Gaussion force statistics influence common ways to calculate fatigue and extreme loads. Berg et al. [15] reported a vanishing effect of non-Gaussion turbulence on extreme and fatigue loads based on an LES wind field in combination with HAWC2 [16]. However, in numeric studies the challenge is to generate synthetic wind field featuring correct statistics of both, velocity increments and velocity values. At ForWind, we use the Continuous Time Random Walk (CTRW) model, which is know e.g. from electron transport and molecular movement, in combination with LES to generate synthetic wind fields. An early version of this approach was used in [10], showing insignificant effects. Recent improvements allow for a more realistic generation of synthetic wind fields in the sense of one- and two-point statistics. Those wind fields were used by Schwarz et al. [17] in combination with a Blade Element Momentum approach and the NREL 5MW reference turbine in order to quantify the effects of non-Gaussian velocity increments on fatigue load calculations. Figure 1 shows the equivalent fatigue loads based on a rainflow counting. Results should be seen as preliminary and are taken from [17].
 Clearly, the inflow conditions featuring intermittent velocity increments result in increased fatigue loads relative to the reference case featuring Gaussian statistics.

Summarizing, we believe that the non-Gaussian character of atmospheric velocity increments, on time scales affecting the rotor, do impact loads of wind turbines. However, it is important to notice that it is today not clear and a current research question, how intermittency affects common ways of load calculations (rainflow counting for example). This possibly strongly depends on details such as time scales etc. Proper numeric and experimental tools for investigations are being developed and so-

[Figure]

**Figure 1.** Effective fatigue loads, absolute (left) and relative (right) of the NREL 5MW reference turbine exposed to Gaussion and non-Gaussian wind fields generated with the CTRW model. Taken from [17].

phisticated studies are limited. Therefore, a complete and conclusive answer is not within the scope of this manuscript. Nevertheless, we do agree that this should be stated more clearly in the manuscript and that it should be elaborated in more detail. We suggest to update the discussion section as follows:

p.15, ll. 5 ff:
*This becomes important when assessing the applicability of active wake steering approaches, as a gain in power has to be balanced with a potential load increase, affecting maintenance costs and the lifetime of turbines overall.*
*It should be noted that it is to date not clear to what extent high TKE levels and intermittent force data are affecting common ways of fatigue and extreme load calculations. This important aspects needs to be addressed in future works. Possibly, it strongly dependents on details such as considered time scales. In our opinion, it is likely that non-Gaussian inflow is linked to drive train, gear box or pitch systems failures, especially because those inflow characteristics are not accounted for in standard models used in the design process. The velocity deficit [...]*

**In the companion paper, figure 11 shows a reduction in TKE during wake steering. If one is considering wake steering, to what extent would a reduction in TKE counter-balance a change in increment velocity? Is there a method to weigh these two changes?**
We believe that a quantification of the impact of the inflow's TKE on e.g. fatigue loads of a turbine is a challenging tasks. To our knowledge a direct method is yet to be found. The same holds for intermittency. Thus, there is not a method to weigh both flow situations in terms of loads quantitatively. However, we do agree that some speculation about these questions can improve the discussion section of the manuscript. Please refer to the above changes (p.15,ll 5 ff).

Further Details:

It is a well-known feature of the atmospheric boundary layer that velocity increments (time scale: order $\sim$ seconds) feature non-Gaussian characteristics. This has been summarized in [1], Figure 2 of this reply shows a screen shot. In the wind energy conlinked to torque fluctuations (e.g., Musial et al., 2007; Feng et al., 2013). Next, turbulent wind affects extreme and fatigue loads, which is clearly related to the lifetime of WECs (Burton et al., 2001).

Wind dynamics in the atmospheric boundary layer have been investigated extensively. Here, one has to differentiate between analyses concerning the statistics of the wind speed *values* and velocity *increments*. The wind velocities might become anomalously distributed due to large-scale meteorological events like downbursts or thunderstorms (De Gaetano et al., 2014). Velocity increments, on the other hand, statistically characterize the temporal aspect of fluctuations, whose non-Gaussian statistics are well-known from small-scale turbulence (Frisch, 1995). Active systems, like wind turbines discussed here, adapt to actual wind situations. Thus, in this paper we focus on wind speed changes within seconds, i.e., by the corresponding increments. Numerous studies have reported on non-Gaussian characteristics of wind speed increments; see, e.g., Boettcher et al. (2003), Liu et al. (2010), Morales et al. (2012), and Wächter et al. (2012). Furthermore, findings of non-Gaussian wind statistics have been implemented in simulations by a variety of methods; see, e.g., Nielsen et al. (2007), Mücke et al. (2011), and Gong and Chen (2014).

**Figure 2.** Screen shot taken from [1]. The highlightes references are [2, 3, 4, 5, 6].

text, this is of particular interest because those characteristics are not implemented in standard wind field models such as the Kaimal model, which is suggested to be used in the design process by the norm IEC 61400-1. Figure 3 shows a screen shot taken from [1], showing distributions of velocity increments of two time series, one is based on offshore measurements in the north sea (FINO1 measurement platform), the other one is based on a synthetic wind field based on the Kaimal model [7], generated in TurbSim [8]. Both time series are equal regarding mean values and turbulence intensity, however, as the graph shows, the distributions of velocity increments are not grasped correctly by the Kaimal model, which features purely Gaussian statistics. So far, it is clear that atmospheric wind features non-Gaussian increment statistics on small scales.

**Table 1.** First two statistical moments and turbulence intensities of a synthetic wind speed time series based on the Kaimal model and offshore data (FINO1). Values are rounded to two decimal places.

| Time series | $\langle u \rangle$ [m s$^{-1}$] | $\sigma_u$ [m s$^{-1}$] | TI [%] |
|---|---|---|---|
| Kaimal | 7.51 | 0.54 | 7.21 |
| FINO1 | 7.50 | 0.54 | 7.18 |

[Figure]

**Figure 1.** $p(u_\tau)$ for data sets based on the Kaimal model (dashed black line) and for offshore measurements, conditioned so that $\langle u \rangle = 7.5 \pm 0.5$ m s$^{-1}$ (solid black). The PDFs for each scale are shifted vertically for better comparison, which is done throughout this paper. Scales from top to bottom $\tau = \{1, 5, 10, 30, 60$ s$\}$.

**Figure 3.** Screenshot taken from [1]. FINO1 refers to offshore measurement data, Kaimal is a synthetic wind field based on the Kaimal model, generated by TurbSim.

**2)**

**Could the authors elaborate further on the connections to the companion paper. Would it make sense to bring the TKE analysis of the companion paper into this paper, and move the analysis of wake position to the companion paper? Feel free to reject this suggestion if I misunderstand the distinctions between the papers, My thinking is just that, for example, if only one of the papers dealt with estimating wake position, then this could make each of the papers more focused on specific effects. But, it would also be acceptable to further elaborate on the focus of the two papers, where they overlap and where they diverge.**

Thank you very much for pointing this out and adding these constructive ideas to the discussion. Generally, the idea of dividing both manuscripts is the following: This paper here compares both turbines. Therefore, the turbine is the changing variable and we limited examined cases to one downstream distance ($6D$) and one inflow condition (uniform turbulence/grid). Comparing data of 2 turbines, 3 yaw angles, 2 distances and multiple inflow conditions would simply be too much for one manuscript. The means of comparison are the velocity deficit, the TKE and the intermittency parameter $\lambda^2$. The companion paper focuses on the impact of different inflow conditions. Therefore, the changing variable is the turbulence grid (no grid, uniform grid, shear grid) and therewith the inflow. The turbine was limited to one turbine only to keep the focus. Although both papers investigate the TKE in the wake, one main point in this manuscript is how findings are different/similar regarding both turbines, while the main point in the companion paper is how the findings change with different inflow conditions. Because of that, we would like to keep the distinctions as done in the discussion papers. However, we think it adds clarity to mention parts of this discussion in the introduction and suggest to reformulate more clearly:

p. 3, ll. 1 ff.
*This work is part of a joint experimental campaign by the NTNU in Trondheim and ForWind in Oldenburg.  While this paper compares the wakes behind two different model wind turbines during one inflow condition, a second paper by [18] examines the influence of varying inflow conditions on the wake of one model wind turbine.*

**3)**

**Finally, the difference in rotation direction between the turbine models is very interesting. The authors use this difference to explain the asymmetries in vertical transport and tilt, could it also explain differences in displacement for positive vs negative yaw observed in the companion paper? Does the size of observed vortices vary with whether the vortex shed by misalignment is rotating in the same direction as the wake?**

Thank you very much for bringing up this interesting aspect. To address the first question, Figure 4 of this document shows the results of the wake center quantification as proposed in Section 2.2 of the manuscript. The figure is the basis of Table 2 of the

manuscript. As Table 2 of the manuscript and Figure 4 of this document show, the same deflection magnitude for either direction of yaw misalignment was found for the ForWind turbine. Differences occur behind the NTNU turbine only using the method described in Section 2.2 of the manuscript. Therefore, we cannot conclude with certainty that the direction of rotation is the reason for asymmetric deflections. If that hypothesis held, one would expect that the deflections behind the ForWind turbine would be asymmetric as well but the other way round, which it is not.

Consequently, one can only speculate about the reasons for the distinctions between the turbines in terms of asymmetric deflection regarding $\gamma = \pm 30°$. Intuitively, one would assume reasons are connected to the differences amongst the turbines, being:

- blockage

- geometry (tower, nacelle)

- rotor (airfoil, rotor tips,...)

In my opinion it is important that blockage/wind tunnel effect are more influential using the NTNU turbine, especially during yaw misalignment with a wake deflection. Wind tunnel effects might play a role regarding the distinctions between both turbines shown in Figure 4 of this document.

However, the data does not allow a certain reasoning, so its all a bit speculative.

[Figure]

**Figure 4.** Potential power $P^*$ as described in Section 2.2 of the manuscript for varying horizontal positions $z$. $x/D = 6$.

At the same, a "wake center" is somewhat a vague term. We use the method of a potential downstream turbine's power because we believe it is closest to the potential application of wake deflection studies. We only considered variations in $z$ direction, which should be kept in mind.

Regarding the second question, I think this is a very interesting train of thought. Ideally, one would have to consider all three flow components for a proper interpretation of the evolving vortex pairs. However, only two components were recorded for the majority of this campaign. Those ($u(t)$ and $v(t)$) are shown in Figures 5 and 6 of this document for the ForWind turbine and the NTNU turbine, respectively. Showing both turbines (and thus both rotational directions), both yaw angles and both flow components, one can compare as much as the data allows. However, as the third flow component was not recorded, some speculation about the vortex pair is probably inevitable. Starting with the ForWind turbine (Fig. 5 of this document), we believe the plots show quite symmetric situations comparing positive and negative yaw misalignment. Confirming Table 2 of the manuscript, also the $v$ component shows very symmetric contours, regarding position, shape and magnitude of the dipoles. One expects a strong horizontal velocity component at hub height towards positive $z$ direction for $\gamma = -30°$ and in negative $z$ direction for $\gamma = +30°$, resulting in two counter rotating vortex pairs (cf. Fig.6 of the companion paper). As the contours for $\gamma = -30°$ and $\gamma = +30°$ look very symmetric regarding $u(t)$ and $v(t)$, one cannot conclude that the shed vortices are much different regarding the direction of yaw misalignment.

Looking at Figure 6 of this document, contours behind the NTNU rotor are slightly asymmetric, which is expected based on Figure 4 of this document.

I believe, similarly as for the first question, one has to think about the differences listed above and some speculation is inevitable. To me, it is more likely that those asymmetries are caused by wind tunnel/blockage effects. Consequently, we do not believe that there is a clear connection between the size of the vortex pair and the direction of yawing / the direction of rotation.

[Figure]

**Figure 5.** Wakes behind the ForWind turbine at $x/D = 6$. Left column: $\gamma = -30°$, right column: $\gamma = +30°$. Top row: $\langle u \rangle / u_{ref}$, bottom row: $\langle v \rangle / u_{ref}$.

[Figure]

**Figure 6.** Wakes behind the NTNU turbine at $x/D = 6$. Left column: $\gamma = -30°$, right column: $\gamma = +30°$. Top row: $\langle u \rangle / u_{ref}$, bottom row: $\langle v \rangle / u_{ref}$.

**References**

[1] Schottler, J., Reinke, N., Hölling, A., Whale, J., Peinke, J., and Hölling, M., "On the impact of non-Gaussian wind statistics on wind turbines – an experimental approach," *Wind Energy Science*, Vol. 2, No. 1, jan 2017, pp. 1–13.

[2] Frisch, U., *Turbulence : the legacy of A.N. Kolmogorov*, Vol. 1, Cambridge university press, 1995.

[3] Boettcher, F., Renner, C., Waldl, H. P., and Peinke, J., "On the statistics of wind gusts," *Boundary-Layer Meteorology*, Vol. 108, No. 1, 2003, pp. 163–173.

[4] Liu, L., Hu, F., Cheng, X.-L., and Song, L.-L., "Probability Density Functions of Velocity Increments in the Atmospheric Boundary Layer," *Boundary-Layer Meteorology*, Vol. 134, No. 2, 2010, pp. 243–255.

[5] Morales, A., Wächter, M., and Peinke, J., "Characterization of wind turbulence by higher-order statistics," *Wind Energy*, Vol. 15, No. 3, 2012, pp. 391–406.

[6] Wächter, M., Heißelmann, H., Hölling, M., Morales, A., Milan, P., Mücke, T., Peinke, J., Reinke, N., and Rinn, P., "The turbulent nature of the atmospheric boundary layer and its impact on the wind energy conversion process," *Journal of Turbulence*, Vol. 13, 2012, pp. N26.

[7] Kaimal, J. C. J., Wyngaard, J. C. J., Izumi, Y., Coté, O. R., and Cote, O. R., "Spectral Characteristics of Surface-Layer Turbulence," *Quarterly Journal of the . . .*, Vol. 98, No. 417, 1972, pp. 563–589.

[8] Jonkman, B. J., "TurbSim user's guide: Version 1.50," 2009.

[9] Milan, P., Wächter, M., and Peinke, J., "Turbulent character of wind energy," *Physical Review Letters*, Vol. 110, No. 13, 2013, pp. 1–5.

[10] Mücke, T., Kleinhans, D., and Peinke, J., "Atmospheric turbulence and its influence on the alternating loads on wind turbines," *Wind Energy*, Vol. 14, No. 2, 2011, pp. 301–316.

[11] Jonkman, J. M. and Buhl Jr, M. L., "FAST user's guide," *National Renewable Energy Laboratory, Golden, CO, Technical Report No. NREL/EL-500-38230*, 2005.

[12] Moriarty, P. J. and Hansen, A. C., *AeroDyn theory manual*, Citeseer, 2005.

[13] Tavner, P., Qiu, Y., Korogiannos, A., and Feng, Y., "the Correlation Between Wind Turbine Turbulence and Pitch Failure," *Euro. Wind Energy Conf.*, 2011, pp. 2–6.

[14] van Kuik, G. A. M., Peinke, J., Nijssen, R., Lekou, D., Mann, J., Sørensen, J. N., Ferreira, C., van Wingerden, J. W., Schlipf, D., Gebraad, P., Polinder, H., Abrahamsen, A., van Bussel, G. J. W., Sørensen, J. D., Tavner, P., Bottasso, C. L., Muskulus, M., Matha, D., Lindeboom, H. J., Degraer, S., Kramer, O., Lehnhoff, S., Sonnenschein, M., Sørensen, P. E., Künneke, R. W., Morthorst, P. E., and Skytte, K., "Long-term research challenges in wind energy – a research agenda by the European Academy of Wind Energy," *Wind Energy Science*, Vol. 1, No. 1, 2016, pp. 1–39.

[15] Berg, J., Natarajan, A., Mann, J., and Patton, E. G., "Gaussian vs non-Gaussian turbulence: impact on wind turbine loads," *Wind Energy*, Vol. 17, No. April 2013, 2016, pp. n/a–n/a.

[16] Larsen, T. J. and Hansen, A. M., *How 2 HAWC2, the user's manual*, Risø National Laboratory, 2007.

[17] Schwarz, C. M., Ehrich, S., Martín, R., and Peinke, J., "Fatigue load estimations of intermittent wind dynamics based on a Blade Element Momentum method," *The Science of Making Torque from Wind 2018 (to be submitted)*, 2018.

[18] Bartl, J., Mühle, F., Schottler, J., Sætran, L., Peinke, J., Adaramola, M., and Hölling, M., "Wind tunnel experiments on wind turbine wakes in yaw: Effects of inflow turbulence and shear," *Wind Energy Science Discussions*, , No. January, jan 2018, pp. 1–22.

---

## Author Comment (AC2) · 2 Mar 2018

Authors' response to Anonymous Referee #2:

We, the authors, are very thankful for the detailed and constructive comments and greatly appreciate the willingness to review our manuscript. Please find our responses below. The original comments are shown in **bold** with the respective answers below. Excerpts of the manuscript are shown in *italic writing*, whereas additions are written in blue and deleted parts in red.
Please note that the format of citations in manuscript excerpts might be changed.
Thank you very much for your efforts,

Jannik Schottler on behalf of all authors
* * *
**1)**
**Although the title clearly mentions the paper deals with a wind tunnel test, it would be good to exercise some caution in the text on the application of the results to the 'real world'.**

Thank you very much for this valuable input to the discussion. We do agree that it is important discuss real life application of the findings and want to adapt the manuscript accordingly.
Generally the (scientific) interest in wind turbine wakes is closely related to the 'real world' as wake effects are known to cause an increase in the cost of energy. Therefore, a mitigation of wake effects would be economically beneficial for wind farm operators and turbine manufacturers. As described in the introduction, wake steering through intentional yaw misalignment is one concept potentially capable of mitigating wake effects in wind farms, however, prior to applying the concept, the effects have to be studied carefully in numeric simulations, experimentally and in field tests, all of which are currently done. Going towards the concrete finding of this study that are summarized in the conclusion, we think the formation of a curled wake during yaw misalignment is important when assessing the applicability of wake steering concept. Those flow conditions become inflow conditions for downstream turbines, therefore an inhomogeneous flow field is an important feature for resulting loads which need to be investigated when judging active wake steering methods. Similarly, the curled shape shows that line measurements at hub height are not sufficient when quantifying wake deflection magnitudes. This is stated in p.15, ll. 6-8 in the manuscript.

Next, a ring of super-Gaussian velocity increment surrounding the velocity deficit of a wake, thus having a significantly larger diameter than the rotor, is one main finding of this paper. The importance of those statistical characteristics are potential load increases. For a more detailed elaboration, please refer to comment/answer #1 of the Referee #1. In a 'real world' scenario, the applications are two folded:

- In wind farm layout optimization, the width of of a wake is a crucial parameter, especially for lateral turbine spacing. Our results suggest the width of a wake significantly increases when taking two-point quantities into account (here: $\lambda^2$). Exemplary, in a (laterally) densely spaced wind farm, a turbine might operate in free stream condition considering the velocity deficit, but might be affected

by the ring of high $\lambda^2$ values shown in Fig. 7 of the manuscript. This difference becomes more clear looking at Fig. 10. Power *and* loads are being considered when optimizing a layout, loads are potentially strongly affected by the findings of the paper.

- As stated in the introduction, wake steering approaches through yaw misalignment are heavily discussed in the research community. The idea is to steer a wake away from a potential downstream turbine to mitigate power losses through wakes, thus gaining net power. Just as in layout optimizations, not only power but also loads have to be considered, which again might be affected by our findings: Looking at Fig. 13 for example at $z = -0.5, y = 0$ a potential in-line downstream turbine would experience more free stream velocity, thus a power increase. Taking two-point statistics into account however shows that the exact same location would experience flows featuring high $\lambda^2$ values. Please refer to comment #1 of referee #1 for a more detailed elaboration about the connection between loads and intermittency.

We suggest to formulate more clearly in the updated discussion section:

p.14, ll.8 ff:
*Consequently, our findings should be considered in wind farm layout optimization approaches, where a wake's width  is a crucial parameter for radial turbine spacing. As layouts are being optimized regarding power and loads, the latter might be significantly affected by taking into account intermittency and the resulting increased wake width. Possibly, the ring of non-Gaussian velocity increments [...].*
* * *
**2)**
**The reported high thrust coefficients corresponds to rather high axial induction factors towards the turbulent wake state, in how far is this representative for real life turbines nowadays and how would this affect the observed wake shapes? Has there been any attempt to clarify the effect of operational conditions on observations (partial load. full load)**

Thank you very much to pointing the attention to the high thrust for the ForWind turbine. We noticed a non-consistency here and want to correct it: Regarding Table 1 of the manuscript, the thrust coefficient of the NTNU turbine was calculated with subtracting the thrust caused by the tower. For the ForWind turbine, the value is based on the total turbine thrust, including the tower structure. This should be corrected in the manuscript and clearly stated. We apply the following correction of the ForWind thrust coefficient:

The tower structure of the ForWind turbine is simplified as a cylindrical structure of $4\,\mathrm{cm}$ diameter. At the inflow velocity of $\langle u(t) \rangle = 7.5 ms^{-1}$, the resulting Reynolds number is $Re_{tower} \approx 2.1 \times 10^4$. Based on Figure 1, the resulting drag coefficient of a circular cylinder and thus the tower is

$$c_{T,tower} \approx 1.2. \tag{1}$$

With the thrust on the tower being

$$F_{T,tower} = 2c_{T,tower}/u^2 \, \rho \, A_{tower}, \tag{2}$$

we can now calculate the corrected thrust coefficient to be

$$c_T^* = 2(F_{tot} - F_{tower})/u^2 \rho A_{rotor} \tag{3}$$

$$c_T^* \approx 0.87. \tag{4}$$

Therewith, the thrust coefficient is the same for both turbines. We want to correct this is in the manuscript as follows:

p. 3, Table 1:

Table 1: Summary of main turbine characteristics. The tip speed ratio (TSR) is based on the free stream velocity $u_{ref}$ at hub height. The Reynolds number at the blade tip, $Re_{tip}$, is based on the chord length at the blade tip and the effective velocity during turbine operation. The blockage corresponds to the ratio of the rotor's swept area to the wind tunnel's cross sectional area. The direction of rotation refers to observing the rotor from upstream, with (c)cw meaning (counter)clockwise. The thrust coefficients were measured at $\gamma = 0°$ and corrected for thrust on the tower and support structure..

| Turbine | Rotor diameter | Hub diameter | Blockage | TSR | $Re_{tip}$ | Rotation | $c_T$ |
|---------|---------------|-------------|----------|-----|-----------|----------|-------|
| ForWind | 0.580 m | 0.077 m | 5.4 % | 6 | $\approx 6.4 \times 10^4$ | cw | 0.87 |
| NTNU | 0.894 m | 0.090 m | 13 % | 6 | $\approx 1.1 \times 10^5$ | ccw | 0.87 |

p.11, l.20:
*In [1], where the same setup was used, the skew angle for the NTNU rotor decreased from $x/D = 3$ to $x/D = 6$, which is a further indication for wall effects due to blockage, especially during yaw misalignment. Furthermore, both values show smaller angles as for the ForWind turbine. *

p.12, l.11:
*As already seen in Figure 11, the wakes behind the ForWind turbine are deflected further and the curled shape is pronounced stronger, which can be attributed to  blockage effects. Figure 12(b) also shows that the wakes behind both turbines are slightly tilted. Looking at the black curves (ForWind turbine), an asymmetry can be noticed as the curves are tilted towards the left, while the red curves are tilted towards the right.*

Barthelmie et al. report a thrust coefficient of $c_T \approx 0.8$ for Siemens 2.3-MW and 2.0MW Vestas V80 turbines. Trujillo et al. show a $c_T$ of 0.77 for Adwen AD 5- 116 turbines, formerly called M5000-116. This list shows a bit more quantitatively, that a value of 0.87 is slightly high, although the theoretical optimum is at $c_T = 8/9 \approx 0.89$ [3]. When discussing the effect on our observations, one has to distinguish between

[Figure]

**Fig. 1.12.** Circular cylinder: drag coefficient vs. Reynolds number
○          measurements by C. Wieselsberger, see H. Schlichting (1982), p. 17
- - - - -    asymptotic formula for Re → 0: $c_D = \frac{8\pi}{\mathrm{Re}}[\triangle - 0.87\,\triangle^3 + \cdots]$,
                with $\triangle = [\ln(7.406/\mathrm{Re})]^{-1}$, $\mathrm{Re} = Vd/\nu$, $c_D = 2D/(\varrho V^2 bd)$
- - - - - ·   numerical results by A.E. Hamielec; J.D. Raal (1969) and
                B. Fornberg (1985) for steady flow
Re = 300:    steady: $c_D = 0.729$, after B. Fornberg (1985)
                unsteady: $c_D = 1.32$, after R. Franke; B. Schönung (1988)

**Figure 1.** Screenshot taken from [2], drag coefficient over Reynolds number for a circular cylinder.

the different findings as done below:

Curles wake during yaw misalignment:
Two further experimental studies report on a curled wake shape during yaw misalignment. Bastankhah & Porté Agel [4] use a small turbine model of $c_T = 0.82$, while Howland et al. [5] use a drag disc of $c_T = 0.75$. Similarly, Berdowski et al. [6] simulated an actuator disc of $c_T = 0.89$. All three studies report the same general effect of a curled wake shape during yaw misalignment. Thus we think, qualitatively the effect does not depend on the thrust coefficient significantly.

Areas of high $\lambda^2$ values surrounding the vel. deficit:
To our knowledge, the ring of intermittent flow structures surrounding a velocity deficit of a wind turbine's wake as shown in Figure 7 of the manuscript, has not been reported before. Therefore, the effect of different thrust coefficients is hard to predict. However, speculating that the picture of the origin as described in the discussion section (p. 14, ll. 9-13) is correct, I would suspect a variation of the thrust coefficient would not affect the qualitative effect significantly. Of course the thrust has to be high enough to create a wake in the first place. In fact, during yaw misalignment the thrust in main flow direction is decreased and we do observe the same effect there, which supports the above speculation.
* * *
**3)**
**Blockage. Referred paper on tunnel effects refers to blockage correction (to correct free stream velocity and modify power and loads). Does the same conclusion hold for measured wake velocities or are they more sensitive to tunnel effects? Is there an influence of the asymmetry of the test section on the measured wake shape at 6D in yaw?**

Thank you for the comment. We assume the referred paper on wind tunnel effects is Chen and Liou (2011) [7]. Unfortunately, it is not really clear which conclusion is meant in the referee comment, we assume the assumption of neglect-able blockage effects for a cross-sectional blockage ratio of $\leq 10\%$ is meant here.
We do believe that our results support the assumption of 10% blockage ratio being a good estimation for neglect-able blockage effects, even for wake velocity measurements 6D downstream. Figures 5 and 11 of the manuscript do not show any speed up effects behind the ForWind turbine (ratio $< 10\%$), which are visible behind the NTNU turbine (ratio $> 10\%$). Further, the wake center position based on the approach described in Section 2.2 of the manuscript result in symmetric values for positive and negative angles of yaw misalignment and slightly asymmetric values for the NTNU turbine. Thus, we conclude that the suggested 10% is a good estimation of blockage effects becoming noticeable. We stated this in the result (p. 7, ll. 26-27) and in the conclusions (p. 15, ll. 25-27) of the manuscript, however we suggest to reformulate more clearly:

*Minor differences could be ascribed to the more prominent blockage (12.8% vs 5.4%) in the NTNU setup, confirming findings by Chen et al. [7] even for wake velocity measurements, who state blockage effects can be neglected for a blockage ratio $\leq 10\%$.*

We assume that 'asymmetric test section' refers to the test section having different extensions in $y$ than in $z$ direction. We believe that during yaw misalignment and the resulting wake deflection in $z$ direction, the tunnel width ($z$ direction) is the parameter potentially affecting the wake extension, especially for the larger rotor as previously discussed. For both directions, larger measures would be of advantage, however, we do not believe that both extensions being asymmetric cause specific effects.
* * *
**4)**
**2.1 pp3 Please state the cause/reason for the different TI. How was the homogeniety verified, do I understand correctly that standard deviation of flow velocity was the same in all three directions??**

We certainly agree that the difference in TI is well worth discussing, thank you very much for pointing it out. The reasons for the different values in inflow turbulence intensity are wind tunnel limitations, unfortunately. The same turbulence grid at the test section inlet was used for both turbines. However, at first, the stream wise position of the smaller ForWind turbine was chosen as a compromise of two aspect: on the one hand, the position should be at a sufficiently large distance from the turbulence grid to allow turbulent mixing. On the other hand, the position should enable a traversing of the LDA system 6 rotor diameters downstream of the turbine.
As the NTNU rotor is larger than the ForWind rotor, the NTNU turbine had to be installed closer to the turbulence grid and therewith to the inlet to the test section, to allow wake measurements 6 rotor diameters downstream of the turbine in the test section of 11.15 m length. The traversing system in the NTNU wind tunnel is permanently installed, so moving the turbine was the only way to access downstream distances of 6D. Consequently, the grid generated turbulence did not decay to the same extent as for the ForWind turbine, unfortunately resulting in different turbulence intensities in the inflow.
Figure 2 of this document shows the resulting values of turbulence intensities,

$$TI := \sigma_u / \langle u \rangle \,, \tag{5}$$

over height, measured at a vertical line at the respective turbine's position, without the turbine being installed. Vertically, deviations in TI are within $\pm 1.7\%$ for the ForWind turbine and within $\pm 3\%$ for the NTNU turbine. Equation 5 states that only the stream wise flow component $u$ was used, as not all three flow components were recorded.

[Figure]

**Figure 2.** Turbulence intensity (TI) of the inflow for both turbines, measured in one vertical line at the turbine position without the turbine being installed.

We suggest to add the information about the turbulence intensity to the caption of Table 1 of the manuscript:

p.3, ll. 16 ff:
*For the NTNU turbine, the reference velocity measured in the empty wind tunnel was $u_{ref,NTNU} = 10\,ms^{-1}$ at a turbulence intensity of $TI = \sigma_u/\langle u \rangle = 0.1$. For the ForWind turbine, the inflow velocity was $u_{ref,ForWind} = 7.5\,ms^{-1}$ and $TI = 0.05$. In both cases,*  *u(t) was homogeneous within $\pm 6\,\%$ and the TI within $\pm 3\%$ on a vertical line at the turbine's position.*

**5)**
**2.2 pp4 motivate choice for x/D=6**
Thank you for this comment. As previously mentioned in the answer to comment #4), six rotor rotor diameters is the upper limit that can be realized at the wind tunnel facility, setting the upper boundary of possible downstream distances. Within the project, we decided to measure two downstream positions to get an insight in downstream wake development. As 6D is the upper limit we chose this distance along with 3D as second distance, which was investigated in previous studies using a comparable setup [8, 9]. This manuscript here focuses on the comparison of both turbines. Therefore, the turbine is the changing variable and we limited examined cases to one downstream distance ($6D$) and one inflow condition (uniform turbulence/grid) as comparing data of 2 turbines, 3 yaw angles, 2 distances and multiple inflow conditions would be too much for one manuscript. In the companion paper (Bartl et al. 2018 [10]), only one turbine was investigated, however, during different inflow conditions and both downstream distances.

| wind farm | Horns Rev 1 | Rødsand | Lillgrund | North Hoyle | Nysted |
|---|---|---|---|---|---|
| **spacing** [D] | 7 | 5.2–7.8 | 3.3–4.4 | 4.4–10 | 5.8–10.4 |

Table 2: Overview of wind farm spacings as stated in [11].

The study by Walker et al. (2016) [11] uses measurement data from five offshore wind farms: Horns Rev 1, Rødsand II, Lillgrund, North Hoyle and Nysted, listed in Table 2. Averaging all values results in $\approx 6.47\,D$ as average turbine spacing. We thus conclude that the (somewhat forced) choice of 6D is a downstream distance relevant to consider.
* * *
**6)**
**5 pp15, does blockage also depend on Ct?**
In this study, we did not investigate how varying the thrust coefficient affects blockage effects on the wake velocities. In our opinion, it would be very hard to isolate the effect of $c_T$ on the blockage effect as varying the $c_T$ would affect the wakes regardless of blockage effect. One study examining blockage effects during wind tunnel experiments using model wind turbines is Chen and Liou (2011) [7], although the focus is on turbine performance rather than wake measurements. Nevertheless, Figure 3 of this document shows that blockage effects (on performance) are dependent on the tip speed ratio. Thus, the $c_T$ should impact blockage effects on performance.

**4. Conclusions**

This research provides quantitative results for the effects of tunnel blockage on the power coefficients of small horizontal-axis wind turbines in wind tunnel tests under different tip speed ratios (TSR), rotor pitch angles ($\beta$), tunnel blockage ratios (BR), and air freestream velocities ($U_\infty$). Results indicate that the tunnel blockage effects and, thus, the blockage factor (BF) are largely dependent on TSR, BR, and $\beta$, and weakly dependent on $U_\infty$. The blockage effects increase as TSR and BR increase, and as $\beta$ decreases. The

**Figure 3.** Screen shot taken from [7].
* * *
**7)**
**5 pp15 It is stated that another paper "Bartl, J., Mühle, F., Schottler, J., Sætran, L., Peinke, J., Adaramola, M., and Hölling, M.: Experiments on wind turbine wakes in yaw: Effects of inflow turbulence and shear, Wind Energy Science, submitted, 2017." discusses the effect of inflow TI. " Since the differences between the measurements on the 2 turbines are discussed in the conclusions, what would be the effect of the different inflow TI for the 2 campaigns on the measured differences?**

Thank you very much for this constructive point. I think to answer this question, one has to distinguish between the different findings/distinctions and discuss them separately as done below:

The manuscript reports different wake deflection magnitudes for both turbines (cf. Table 2 of the manuscript). The companion paper Bartl et al. (2018) [10] discusses differences in the wakes during yaw misalignment for the NTNU turbine and turbulence intensities of about 0.23% and 10%. Figure 4 shows the wakes behind the NTNU turbine for both angles of yaw misalignment and both turbulence intensities. [10] shows detailed elaboration on the differences, some of which I summarize here

[Figure]

**Figure 4.** $\langle u \rangle / u_{ref}$ behind the NTNU turbine at $x/D = 6$. Left column: $\gamma = -30°$, right column: $\gamma = +30°$. Top row: inflow $TI = 0.23\%$, bottom row: inflow $TI = 10.0\%$. The same data was used in [10].

with regard to the referee comment.

In [10], we apply the same method for wake center detection as described in the manuscript. Figure 5 of this document shows a screen shot taken from [10], comparing the wake deflection magnitudes for different inflow conditions, inflow A and B being 0.23% and 10% inflow TI. As further discussed in the companion paper, the different inflows show only very small distinctions regarding wake deflection.

Further, the maximum velocity deficit is pronounced much stronger at low inflow turbulence. This is expected since higher TI enhances turbulent mixing with the free stream and thus wake recovery. Next, the (curled) wake shape appears to be more 'stable'/defined with higher TI. In my opinion, this effect is due to the *very* low TI of 0.23% (top row), and similar distinctions would not be observable comparing 5% and 10% inflow TI using the same turbine. In fact, the result of this manuscript do show a rather smooth shape for both turbines and thus both inflow TI values.

We suggest to point the reader's attention to the companion paper regarding the issue of 2 different inflow TIs by adding to the discussion of the manuscript:

[Figure]

**Figure 9.** Calculated wake deflection $\delta(z/D)$ at $x/D$=3 and $x/D$=6 for three different inflow conditions A, B and C compared to TI-dependent deflection predictions by Bastankhah and Porte-Agél's wake deflection model. Note that a small offset in $x/D$ of the measured values was chosen for better visibility.

**Figure 5.** Screenshot taken from [10], showing the wake deflection for different in flow conditions.

p.15, 20.ff:

*This confirms findings by [12] and [9], reporting an asymmetric power output of a two-turbine case with respect to the upstream turbine's angle of yaw misalignment. One should bear in mind that the inflow turbulence intensities are different regarding both turbines. We want to point out that the influence of inflow turbulence on the wake deflection is studied in [10], showing no significant effects.*

**RING OF HIGH $\lambda^2$ VALUES**

Regarding the ring of high $\lambda^2$, we see a strong influence of the free stream TI on the magnitude of $\lambda^2$ values. As can be seen in Figure 7 of the manuscript, the $\lambda^2$ values within the ring are considerably higher behind the ForWind turbine and thus for lower free stream turbulence. This connection is further supported by Figure 6 of this document, showing $\lambda^2$ contours behind the NTNU turbine for two different inflow conditions (TI=0.23% and TI=10.0%). Notice that the scale is different in both cases, showing that the values of $\lambda^2$ are higher for the low turbulent case, thus supporting the previous statement. Based on those two comparisons, we assume that a larger gradient in TI (or TKE) between wake and free stream leads to higher $\lambda^2$ values and thus more heavy-tailed increment PDFs on scales comparable to the rotor. This also fits to our interpretation that the ring of high $\lambda^2$ values arises from a transition zone, switching between wake state and free stream state, please see p. 14 ll. 9-12 of the manuscript.

[Figure]

**Figure 6.** Shape parameter $\lambda^2$ at $x/D = 6$ behind the NTNU turbine. Left: free stream TI=0.23%, right: free stream TI=10.0%.

**References**

[1] Schottler, J., Mühle, F., Bartl, J., Peinke, J., Adaramola, M. S., Sætran, L., and Hölling, M., "Comparative study on the wake deflection behind yawed wind turbine models," *Journal of Physics: Conference Series*, Vol. 854, may 2017, pp. 012032.

[2] Schlichting, H. and Gersten, K., *Boundary-Layer Theory*, Vol. 102, Springer Berlin Heidelberg, Berlin, Heidelberg, 2017.

[3] Manwell, J. F., McGowan, J. G., and Rogers, A. L., *Wind energy explained: theory, design and application*, 2009.

[4] Bastankhah, M. and Porté-Agel, F., "Experimental and theoretical study of wind turbine wakes in yawed conditions," *Journal of Fluid Mechanics*, Vol. 806, No. 1, nov 2016, pp. 506–541.

[5] Howland, M. F., Bossuyt, J., Martínez-Tossas, L. A., Meyers, J., and Meneveau, C., "Wake structure in actuator disk models of wind turbines in yaw under uniform inflow conditions," *Journal of Renewable and Sustainable Energy*, Vol. 8, No. 4, 2016.

[6] Berdowski, T., "Three-Dimensional Free-Wake Vortex Simulations of an Actuator Disc in Yaw and Tilt," *2018 Wind Energy Symposium*, No. January, American Institute of Aeronautics and Astronautics, Reston, Virginia, jan 2018.

[7] Chen, T. and Liou, L., "Blockage corrections in wind tunnel tests of small horizontal-axis wind turbines," *Experimental Thermal and Fluid Science*, Vol. 35, No. 3, apr 2011, pp. 565–569.

[8] Schottler, J., Hölling, A., Peinke, J., and Hölling, M., "Wind tunnel tests on controllable model wind turbines in yaw," *34th Wind Energy Symposium*, , No. January, 2016, pp. 1523.

[9] Schottler, J., Hölling, A., Peinke, J., and Hölling, M., "Brief Communication : On the influence of vertical wind shear on the combined power output of two model wind turbines in yaw," , No. 2016, 2017, pp. 1–5.

[10] Bartl, J., Mühle, F., Schottler, J., Sætran, L., Peinke, J., Adaramola, M., and Hölling, M., "Wind tunnel experiments on wind turbine wakes in yaw: Effects of inflow turbulence and shear," *Wind Energy Science Discussions*, , No. January, jan 2018, pp. 1–22.

[11] Walker, K., Adams, N., Gribben, B., Gellatly, B., Nygaard, N. G., Henderson, A., Marchante Jimémez, M., Schmidt, S. R., Rodriguez Ruiz, J., Paredes, D., Harrington, G., Connell, N., Peronne, O., Cordoba, M., Housley, P., Cussons, R., Håkansson, M., Knauer, A., and Maguire, E., "An evaluation of the predictive accuracy of wake effects models for offshore wind farms," *Wind Energy*, Vol. 19, No. 5, may 2016, pp. 979–996.

[12] Fleming, P., Gebraad, P. M., Lee, S., Wingerden, J.-W., Johnson, K., Churchfield, M., Michalakes, J., Spalart, P., and Moriarty, P., "Simulation comparison of wake mitigation control strategies for a two-turbine case," *Wind Energy*, 2014.

---

## Author Comment (AC3) · 2 Mar 2018

Authors' response to Anonymous Referee #3:

We, the authors, are very thankful for the detailed and constructive comments and greatly appreciate the willingness to review our manuscript. Please find our responses below. The original comments are shown in **bold** with the respective answers below. Excerpts of the manuscript are shown in *italic writing*, whereas additions are written in blue and deleted parts in red.
Please note that the format of citations in manuscript excerpts might be changed.
Thank you very much for your efforts,

Jannik Schottler on behalf of all authors
* * *
**1)**
**The anonymous referee #2 has commented on the high induction factors and the choice of position of measurement plane, the different TI, Ct and blockage ratios for the two turbines tested, and the possible impact on the measured wake velocities. I understand that the wake effects are more easily studied at high induction factors, relatively close to the rotor, but I also share ref #2's curiosity about how this relates to real wind farms. I suggest a section showing the Ct vs. wind speed curve for a large modern wind turbine, and a few sentences about typical wind turbine spacings in recently built wind farms (along and across the main wind direction).**

Thank you very much for this comment and the suggestions. The agreement with referee #2 shows that this is indeed an aspect that should be further elaborated on in the manuscript. We kindly ask you to refer to comment/answer #2 of referee #2, were we discussed the thrust coefficient, as well as comment/answer #5, where we discuss the choice of 6D and typical turbine spacings.
* * *
**2)**
**The anonymous referee #1 main comment is on the impact of inflow velocity increments on the loads for a wind turbine. I would like to add a few comments on this topic. Figure 5 shows the mean velocity deficit at 6D behind the rotor. As expected, the wake (in terms of velocity deficit) has expanded somewhat, but at y/D and z/D of 1, we have more or less free stream conditions. Figure 6 shows the influence of the rotor in terms of TKE. Again we see that the wake has expanded, but at y/D and z/D of 1, we are almost at free-stream. Figure 7 is intriguing. Although the wake in terms of mean velocity deficit and TKE is hardly present at y/D and z/D of 1, the shape parameter here shows a strong signal, close to the maximum value across the measurement plane. My main comment is that the shape parameter can be high, but the velocity fluctuations may be too small to affect the loads. I therefore appreciate that the authors in the following figures try to present the results in different ways, but in my opinion, some more figures should be added here. In figure 8, the**

probability density functions at the two points are normalized in different ways to be compared with the same Gauss distribution. What is the ratio of velocity increment standard deviations at the two positions? How would a plot look if the results were normalized in the same manner? In figure 9, the velocity increments at the two positions are again normalized with different standard deviations. I would like to see the corresponding plots also normalized with the standard deviation at D/2.

Thank you very much for this very constructive criticism and interest. I do understand that several questions are posed and details asked for in this comment. Nevertheless, I think it makes sense and adds clarity to answer the aspects mentioned in this comment in one answer as they are closely related. I will refer to specific aspects of the comment throughout the answer in bold writing.

To begin with, I think one has to pay attention to the term 'fluctuations'. Often in literature, fluctuations refer to

$$u'(t) = u(t) - \langle u(t) \rangle, \tag{1}$$

see equation 3 of the manuscript. When stating **'the shape parameter can be high, but the velocity fluctuations may be too small to affect the loads'**, we assume that not fluctuations in the sense of Eq. (1) but velocity increments are meant:

$$u_\tau(t) := u(t) - u(t + \tau), \tag{2}$$

which is statistically different as fluctuations are one-points quantities and increments two-point quantities. For a detailed elaboration we refer to Morales et al. [1].

**'In figure 8, the probability density functions at the two points are normalized in different ways to be compared with the same Gauss distribution.'**
This is not entirely correct and we want to clarify: the PDFs $u_\tau$ are indeed normalized by the standard deviation $\sigma_\tau$ and therewith by different values. This is not done to be compared to the same Gaussian. The normalization allows to purely compare the shape of the individual PDFs. The Gaussian is added to guide the eye as normally one is familiar with the Gaussian shape. As mentioned in the manuscript (p.14, ll. 14 ff), $\lambda^2$ is in indicator for a PDFs shape. Because of that, we normalize the PDFs in Fig. 8 to purely compare the shape and thus visualizing what is expressed by $\lambda^2$.
We fully agree and really appreciate the hint, that for a connection to loads, the absolute values of $u_\tau \, [ms^{-1}]$ are much more intuitive. However, we want to clearly distinguish this and order it the following way:

1. we find a ring of high $\lambda^2$ values (Fig. 7). This parameter expresses the *shape* of a pdf.

2. we show 2 exemplary increment PDF, normalized, in order to actually show the shape that is expressed by $\lambda^2$. We believe the shape is best compared by normalizing by the standard deviation

3. Fig. 10 shows that $\lambda^2$ is high where $\langle u \rangle$ would indicate free steam conditions

4. Now, we think it would be nice to show non-normalized plots, but comparing positions within the ring of high $\lambda^2$ values and the *free stream*. That way, we show much more intuitively that strong velocity jumps (increments) in short time scale happen significantly more often in the ring than in the free stream. Thus, we show that the ring is indeed no free stream as (falsely) suggested my defining a wake width by the velocity deficit in the wake. We show that is is significantly different regarding velocity increments, and therewith of importance.

For the above points 1. and 2., we think Fig. 7 and 8 should stay in the manuscript. Point 3. is expressed by Fig. 10. We suggest to add the following plots to bring across point 4.:

In order to comment on the impact on loads, or at least get a feeling for the potential impact, we agree that a non-normalized presentation is very beneficial. Figure 1 of this document shows the increment time series $u_\tau$ in free stream condition (a) and within the ring of high $\lambda^2$ values (b). One can clearly see that jumps exceeding 2.5m/s happen frequently in (b), and are non existent in the free stream. Hereby we show that this radial position of the wake features significantly different flows than the free stream. To show this more clearly, Figure 2 shows the corresponding increment PDFs, $p(u_\tau)$ of the absolute values. Clearly, one sees the same thing as in Fig. 1 (of this document):

[Figure]

**Figure 1.** Time series of increments $u_\tau(t)$ for the positions $y/D = 0.8$, $z/D = 1$ (free stream,a) and $y/D = 0$, $z/D = 1$. The standard deviations $\sigma_\tau$ are indicated in red.

[Figure]

**Figure 2.** $p(u_\tau)$ of the free stream at $y/D = 0.8$, $z/D = 1$ and of $y/D = 0$, $z/D = 1$, exemplary for the ForWind turbine.

We suggest to update the Results-section of the manuscript the following way:

p.9 ll. 4 ff:
For $z = D$, which lies within the ring of large $\lambda^2$ values, $p(u_\tau)$ strongly deviates from a Gaussian, showing a heavy-tailed distribution . Figure 8 further shows $p(u_\tau)$ based on the model proposed by [2]. Those distributions were evaluated based on the $\lambda^2$ values computed by Equation (6) at $z = D$, visualizing exemplary how well the distributions' shapes are grasped by $\lambda^2$. $u_\tau(t)$  behind the ForWind turbine, cf. Figures 7(b) and 8(b).  Our results show that, depending on[...].

p.11, ll. 3ff:
For illustration, the dotted lines in Figure 10 mark the respective locations. It is shown that the radial areas of TKE and $\lambda^2$ can be related in this way to the velocity deficit. _To get a feeling of the impact on potential downstream turbine, Figure *2 of this response* compares $p(u_\tau)$ in absolute terms at a free stream position,$y/D = 0.8$, $z/D = 1$, and at a position featuring high $\lambda^2$ values, $y/D = 0$, $z/D = 1$, exemplary for the ForWind turbine. It becomes clear that velocity increments exceeding $3\mathrm{ms}^{-1}$ occur much more frequent within the ring of high $\lambda^2$ values than in three free stream. Hereby we show that this radial position of the wake features significantly different flows than the free stream. To compare more visually, Figure *1 of this response* shows the corresponding time series $u_\tau(t)$. Clearly, the spiky signature of extreme events become obvious in Figure_

*1(b), confirming that no free stream condition is reached at $z/D = 1$.*

p.14, ll.2 ff:
*We find heavy-tailed distributions of velocity increments in a ring area surrounding the velocity deficit and areas of high TKE in a wind turbine wake. Thus, the definition of a wake width strongly depends on the quantities taken into account as the ring area features significantly different statistics than the free stream. The heavy-tailed distributions are [...]*

We further suggest to delete Figure 9 of the manuscript. I think Figure 1 of this reply is more valuable and both might be a bit too much.
* * *
**3)**
**Caption of Table 1, pg. 3: Is the effective velocity during turbine operation the relative wind speed with respect to the rotor tip? The blade tip of the ForWind turbine looks like it has a rounded shape. Where is the tip chord defined?**

Thank you for pointing out that we should be a bit more precise here. As correctly described, the effective velocity $vel_{eff}$ during operation is the wind speed the airfoil experiences at the tip. Indeed, the tip of the ForWind blades are somewhat round. To account for this we calculated the effective velocity and Reynolds number at $r \approx 96\%$ blade radius $R$ and not at 100%. At $r = 0.96R$, the cord length is $c_{96\%} \approx 20mm$ and we can calculate the Reynolds number:

$$\omega = \lambda u / R \tag{3}$$
$$vel_{rot} = \omega r = \lambda u \cdot r/R \tag{4}$$
$$vel_{rot} = \lambda u \cdot 0.96 \tag{5}$$
$$vel_{eff} = \sqrt{u^2 + vel_{rot}^2} \tag{6}$$
$$vel_{eff} \approx 45.6 m/s. \tag{7}$$
$$\Rightarrow Re \approx 6.42 \times 10^4 \ . \tag{8}$$

I think it is still fair to call it $Re_{tip}$ and suggest to clarify this in the updated manuscript by adding to the caption of Table 1:

p.3, Tab.1 (caption):
*Summary of main turbine characteristics. The tip speed ratio (TSR) is based on the free stream velocity $u_{ref}$ at hub height. The Reynolds number at the blade tip, $Re_{\text{tip}}$, is based on the chord length at the blade tip and the effective velocity during turbine operation. For the ForWind turbine, $0.96R$ was chosen as radial position to account for the rounded blade tips. The blockage corresponds to the ratio of the rotor's swept area to the wind tunnel's cross sectional area. The direction of rotation refers to observing the rotor from upstream, with (c)cw meaning (counter)clockwise. The thrust coefficients were measured at $\gamma = 0°$ and corrected for thrust on the tower and support structure.*
* * *
**4)**
**2.3, page 6: Please mention if measurements support the assumption about vertical vs transversal fluctuations. I assume you mean $\langle w^2 \rangle$ vs. $\langle w^2 \rangle$.**

Thank you for this hint. In fact, we mean $\langle v'(t)^2 \rangle \approx \langle w'(t)^2 \rangle$ so the approximation of the TKE is satisfied:

$$k = 0.5 \left( \langle u'(t)^2 \rangle + \langle v'(t)^2 \rangle + \langle w'(t)^2 \rangle \right) \tag{9}$$
$$\approx 0.5 \left( \langle u'(t)^2 \rangle + 2\langle v'(t)^2 \rangle \right). \tag{10}$$

We did do measurements supporting this is a fair assumption. We suggest to add this information to the manuscript:

p.6, l. 6 ff:
*For briefness, we write $\langle u \rangle$ instead of $\langle u(t) \rangle$. As the third flow component w was not recorded, we assume*  $\langle w'(t)^2 \rangle \approx \langle v'(t)^2 \rangle$ *so that Equation (2) becomes*

$$k = 0.5 \left( \langle u'(t)^2 \rangle + 2\langle v'(t)^2 \rangle \right) \ , \tag{11}$$

*which will be used in further analyses. Measurements where performed validating this approximation.For a thorough analysis[...]*
* * *
**5)**
**Caption, Figure 3: Consider adding something like: For the NTNU turbine, the wind tunnel walls are located at z/D = +-3.03 and y/D = +-2.02. For the ForWind turbine, the wind tunnel walls are located at z/D = +-4.67 and y/D = +- 3.12**

We agree that this is helpful information and will add it as suggested in the updated manuscript. Thank you for the suggestion. We believe a factor 0.5 is missing in the suggested values, so we would like to edit the caption of Fig. 3 as follows:

p. 5, Fig 3(caption):
*Non-dimensional measurement grid behind the rotor for $\gamma = 0°$. The respective contours of the turbines are shown in black (ForWind) and red (NTNU). For the NTNU turbine, the wind tunnel walls are located at $z/D = \pm 1.5$ and $y/D = \pm 1.0$, for the ForWind turbine at $z/D = \pm 2.34$ and $y/D = \pm 1.56$.*
* * *
**6)**
**Caption, Figure 11, pg. 12: Bottom row.**

Thank you for this hint, it will be corrected.
* * *
**7)**
**Caption, Figure 13: The red marks show the approximation of the respective parameter's radial extension based on $\mu \pm 1\sigma$ and $\mu \pm 2\sigma$ as described**

**in Section 3.1. But I see only two red lines, is it at one or two sigma?**

Thank you for pointing this out. In the TKE contour plots (center column), the red lines correspond to $\mu \pm 1\sigma_\mu$ and in the $\lambda^2$ contours (right column), the red lines correspond to $\mu \pm 2\sigma_\mu$. We suggest to state this more clearly in the caption:

p.14, Fig 13, caption:
$\langle u \rangle / u_{ref}$ *(left column), TKE (center column) and $\lambda^2$ (right column) for $\gamma = -30°$ behind the NTNU turbine (top row) and the ForWind turbine (bottom row). The time scale for $\lambda^2$ corresponds to the length scale of the rotor diameter. The red marks show the approximation of the respective parameter's radial extension based on $\mu \pm 1\sigma_u$ (TKE, middle column) and $\mu \pm 2\sigma_u$ ($\lambda^2$, left column) as described in Section3.1.*

**References**

[1] Morales, A., Wächter, M., and Peinke, J., "Characterization of wind turbulence by higher-order statistics," *Wind Energy*, Vol. 15, No. 3, 2012, pp. 391–406.

[2] Castaing, B., Gagne, Y., and Hopfinger, E. J., "Velocity probability density functions of high Reynolds number turbulence," *Physica D: Nonlinear Phenomena*, Vol. 46, No. 2, 1990, pp. 177–200.

---

## Author Response (AR2)

Authors' response to the Associate Editor's comments:

Dear Associate Editor,
thank you very much for the constructive comments on our revised version of the manuscript. Please find our answers below.

**1) The use of written English can be improved. In principle you may rely on the proof reading service that Copernicus provides, but in this case I would like to suggest that you might spar/discuss a little bit with the other authors to come up with improved English. This mainly holds for the text added with the update of the manuscript.**

Thank you for pointing this out, the updated version of the manuscript was read carefully by the authors, paying special attention to the updated sections.

**2) Regarding the content: a modification of the manuscript that should be dealt with better is that it is "too easy" and (in fact more important) also scientifically improper, to "just" add the words "Measurements where performed validating this approximation" (just below Equation 11) without showing that this measurements indeed prove that the two orthogonal components of the TKE perpendicular to the main wind direction are equal. Showing this should either be done by referring to a scientific publication in which the measurements are published, that then can be quoted, giving also the quantitative proof of the agreement, or by adding a graph (made from the measurements) in the present manuscript that demonstrates that these TKE components are indeed equal.**

Thank you for the hint. We agree that the calculation of the turbulent kinetic energy (TKE) based on only two flow components should be discussed more than what is stated in the revised manuscript.
For one wake, we recorded all three flow components by traversing the measurement plane twice, once recording $u$ and $v$, once $u$ and $w$. That allows us to say a bit more about the assumption.
For that one wake, we can now compare the actual TKE based on all three flow components,
$$k = 0.5 \left( \langle u'(t)^2 \rangle + \langle v'(t)^2 \rangle + \langle w'(t)^2 \rangle \right), \tag{1}$$
with the approximation using only $u$ and $v$ that are available for the majority of wakes considered,
$$k^* = 0.5 \left( \langle u'(t)^2 \rangle + 2\langle v'(t)^2 \rangle \right). \tag{2}$$
As stated below Eq. 5 of the manuscript, that implies the assumption
$$\langle v'(t)^2 \rangle \approx \langle w'(t)^2 \rangle. \tag{3}$$

In Figure 1 of this reply, we show the contour plot of $k$ (left, Eq.1) and compare it

[Figure]

**Figure 1.** Contours of $k$ according to Eq. (1) (left), contour of $k^*$ according to Eq. (2) (middle) and $\langle v'(t)^2 \rangle$ versus $\langle w'(t)^2 \rangle$ (right) for all 357 measurement positions. r is Pearson's correlation coefficient. Data were recorded 6D behind the NTNU turbine at $\gamma = 30°$.

to the contours of $k^*$ (right, Eq. 2). Additionally, the right plot in Figure 1 shows $\langle v'(t)^2 \rangle$ plotted against $\langle w'(t)^2 \rangle$, $R_P$ is the Pearson correlation coefficient.

The contours of $k$ and $k^*$ show that the differences are, considering the depth of our analysis in this paper, indeed neglectable. Further, the right plot in Figure 1 supports the assumption with a high correlation coefficient of $R_P = 0.94$.

We suggest to add this figure to the discussion section of the manuscript, and refer to it in Sec. 2.3. We further distinguish between $k$ and $k^*$ in Section 2.3. Amongst Figure 1 of this document, we add to the discussion:

p.15, ll. 11ff:

*The turbulence intensity in the  wakes revealed very comparable results as the TKE, which is why we restrict our analyses to the TKE. For the majority of wakes considered, only two flow components were recored. For one exemplary wake, however, all three components are available, allowing to examine the assumption of $\langle v'(t)^2 \rangle \approx \langle w'(t)^2 \rangle$, cf. Equation (4). Figure 15 shows the contours of $k$ and $k^*$ as well as $\langle v'(t)^2 \rangle$ versus $\langle w'(t)^2 \rangle$ for all measurement positions. Both contour plots show neglectable differences, confirming the approximation. This is further supported by a high correlation coefficient of 0.94 between $\langle v'(t)^2 \rangle$ and $\langle w'(t)^2 \rangle$.*
*Besides the lateral deflection, [...]*

Further, we want to notify the Editor that there was a slight mistake in Figure 12(b) of the original manuscript: The axes were y/R and y/R (R being the rotor radius), but were falsely labeled y/D and z/D. This was corrected, for consistency the axes are in y/D and z/D now and labeled correctly.

We want to thank the Associate Editor as well as the three referees for the efforts and constructive comments.

Jannik Schottler on behalf of all authors

[revised manuscript text omitted]